# On the Interplay between Graph Structure and Learning Algorithms in Graph Neural Networks

**Junwei Su** [1]   **Chuan Wu** [1]

## Abstract

This paper studies the interplay between learning algorithms and graph structure for graph neural networks (GNNs). Existing theoretical studies on the learning dynamics of GNNs primarily focus on the convergence rates of learning algorithms under the interpolation regime (noise-free) and offer only a crude connection between these dynamics and the actual graph structure (e.g., maximum degree). This paper aims to bridge this gap by investigating the excessive risk (generalization performance) of learning algorithms in GNNs within the generalization regime (with noise). Specifically, we extend the conventional settings from the learning theory literature to the context of GNNs and examine how graph structure influences the performance of learning algorithms such as stochastic gradient descent (SGD) and Ridge regression. Our study makes several key contributions toward understanding the interplay between graph structure and learning in GNNs. First, we derive the excess risk profiles of SGD and Ridge regression in GNNs and connect these profiles to the graph structure through spectral graph theory. With this established framework, we further explore how different graph structures (regular vs. power-law) impact the performance of these algorithms through comparative analysis. Additionally, we extend our analysis to multi-layer linear GNNs, revealing an increasing non-isotropic effect on the excess risk profile, thereby offering new insights into the over-smoothing issue in GNNs from the perspective of learning algorithms. Our empirical results align with our theoretical predictions, *collectively showcasing a coupling relation among graph structure, GNNs and learning algorithms, and providing insights on GNN algorithm design and selection in practice.*

## 1. Introduction

Graph structure data is ubiquitous in the real world and many important learning problems are naturally modelled as a graph. Graph neural networks (GNNs) have emerged as a dominant class of machine learning models specifically designed for learning problems in graph-structured data. They have demonstrated considerable success in addressing a wide range of graph-related problems in various domains such as chemistry (Gilmer et al., 2017; Reiser et al., 2022), biology (Tsubaki et al., 2019; Réau et al., 2023), social networking (Chen et al., 2017; Sheng et al., 2024; Li et al., 2017; Su et al., 2024), and computer vision (Zhu et al., 2022; Yang et al., 2022; Lu et al., 2016; Xu et al., 2017; Zellers et al., 2018). A defining characteristic of GNNs is their use of a spatial approach through message passing on the graph structure for feature aggregation. This enables GNNs to preserve structural information and dependencies from the underlying graph structure, allowing them to be highly effective in tasks such as node regression.

Because of their central role in numerous important applications, there is a growing body of literature on theoretical research on GNNs (see Sec. 2 for a more detailed discussion). Existing theoretical studies on GNNs primarily concentrate on two aspects: their expressive power (Xu et al., 2018) and generalization capabilities under different measures. The expressive power of GNNs refers to their ability to distinguish between different graph structures and effectively capture node relationships and graph topology. Generalization capabilities are often explored through complexity measures, such as VC-dimension (Scarselli et al., 2018) and Rademacher complexity (Lv, 2021), or through information-theoretic measures, like mutual information and entropy. These results provide interesting insight into how powerful GNNs are as a neural model.

**Existing Gap.** *Nevertheless, there is a notable gap in understanding the interplay between learning algorithms (e.g., stochastic gradient descent (SGD)) and graph structure, especially when concerning their generalization performance (excessive risk) in the presence of noise (inter-*

[1]School of Computing and Data Science, University of Hong Kong. Correspondence to: Junwei Su <junweisu@connect.hku.hk>, Chuan Wu <cwu@cs.hku.hk>.

*Proceedings of the 42$^{nd}$ International Conference on Machine Learning*, Vancouver, Canada. PMLR 267, 2025. Copyright 2025 by the author(s).

*polation regime).* There are few studies concerning the behaviour of learning algorithms in GNNs (Awasthi et al., 2021). These studies are limited in two aspects: 1) they are only concerned with the convergence rates of the learning within the interpolation regime (noise-free), and 2) they provide only a very crude connection to the graph structure, typically represented by the maximum or minimum degree, which offers limited insight into the structure of the graph. Recognizing this gap in the theoretical understanding of GNNs, this paper investigates the interplay between graph structure and learning algorithms in the generalization regime (in the presence of noise). We extend standard settings (least squares) from learning theory literature to the case of GNN and aim to answer the following research questions:

*Can graph structure affect the generalization performance of learning algorithms in GNNs? If so, how does the graph structure affect the learning algorithms?*

**Contribution.** The primary objective of this paper is to examine the interplay between learning algorithms and GNNs, particularly focusing on how graph structure impacts the generalization performance (excessive risk) of learning algorithms in the interpolation regime. The main challenges addressed in this research are establishing a connection between graph structure and learning algorithm performance, and creating a robust comparison framework. Our analysis focuses on two central learning algorithms, SGD and Ridge, aiming to understand the influence of graph structure by comparing the performance of these algorithms in different graph types (power-law vs. regular). The contributions and results of this study are highlighted as follows:

1. We extend the existing excessive risk analysis to the context of GNNs, broadening the understanding of these learning algorithms within the learning theory literature. Specifically, we have derived the excessive risk (generalization performance) profiles, including both upper and lower bounds, of learning algorithms (SGD and Ridge) in GNNs (Theorems 4.1 and 4.2). These profiles establish a link between the graph matrix and the excessive risk of learning algorithms in GNNs. They lay the groundwork for our subsequent investigation into the impact of graph structure on the performance of learning algorithms.

2. Based on the established connection between the graph matrix and the excessive risk of different learning algorithms, we further utilize spectral graph theory to link the graph structure with the graph matrix. Through this connection and the results derived, we examine the interplay between graph structure and learning algorithms in GNNs. Specifically, we focus on the two types of connectivity spectra of the graph (regular vs.

power-law) and demonstrate that 1) the excess risk profile of SGD is more favourable (perform better) than Ridge when the underlying graph structure is power-law and 2) the excess risk profile of Ridge is more favourable than SGD when the underlying graph structure is regular (Theorem 4.3). These findings offer practical guidance for learning algorithm selection, suggesting that one should choose a learning algorithm (SGD-like vs. Ridge-like) based on the graph structure.

3. We extend the analysis to the context of multi-layer linearized GNNs. We demonstrate that when increasing GNN layers on power-law graphs, the performance of the learning algorithms exhibits an increasingly non-isotropic effect on the excess risk profile of learning algorithms (Proposition 4.5). Specifically, it becomes easier for the learning algorithms to learn the ground truth in the head eigenspace, while it becomes more challenging in the tail eigenspace. These results provide a new perspective on the well-documented over-smoothing issue in GNNs and offer a surprising insight. While it is commonly understood that adding more layers to GNNs can lead to degraded performance due to over-smoothing, our analysis suggests that increasing the number of layers can be beneficial for learning algorithms such as SGD when the ground truth is concentrated in the head eigenspace.

The empirical results from our controlled experiments with synthetic graph models are consistent with our theoretical predictions, thus validating our analysis and findings. These results not only deepen the theoretical understanding of GNNs but also offer practical insights and guidance for selecting and designing learning algorithms for GNNs based on graph structure.

## 2. Related Work

In this section, we provide a brief overview of the following key aspects of this paper: 1) theoretical studies of GNNs, 2) excessive risk of learning algorithms and 3) spectral graph theory. A more comprehensive discussion of related works is available in the appendix.

**Theoretical Understanding of GNNs.** Due to the empirical success of GNNs, there is a growing body of theoretical studies (see (Jegelka, 2022) for a survey) focusing on the expressive power of GNNs (Sato, 2020; Zhang et al., 2023a;b;c; Xu et al., 2018) and their generalization capabilities. The expressive power of a GNN denotes its ability to distinguish between different graph structures and capture the intricacies of node relationships and graph topology. It is often evaluated by comparing the GNN's discriminative ability against classical graph isomorphism tests, such as



Figure 1. Illustration of how our framework connects graph structure and the performance of learning algorithms. Spectral graph theory connects graph structure and the eigenspectrum of graph matrix. The excessive risk profile of the learning algorithms connects the eigenspectrum of the graph matrix and the performance of the learning algorithms. As such, we establish a framework that can study the interplay between graph structure and learning algorithms in GNNs.

the Weisfeiler-Lehman test (Leman & Weisfeiler, 1968). Meanwhile, generalization studies of GNNs explore how well these models perform on unseen data, utilizing frameworks such as complexity measures (like VC-dimension and Rademacher complexity) (Lv, 2021; Ma et al., 2021; Baranwal et al., 2021; Liao et al., 2021), Neural Tangent Kernel (NTK) (Du et al., 2019) and information-theoretic approaches (such as mutual information and entropy) (Verma & Zhang, 2019; Zhu et al., 2021). These studies seek to determine how factors such as network architecture and properties of the input graphs affect the model's generalization from training to testing data. Therefore, they are orthogonal to the research objectives of this paper.

There is also research on convergence analysis of GNN learning algorithms (Chen et al., 2017; Huang et al., 2018; Chen et al., 2018; Awasthi et al., 2021; Li et al., 2018; Oono & Suzuki, 2020). These studies only examine the convergence rate of the algorithms and are confined to the interpolation regime (noise-free). Their results only provide a crude connection between graph structure (e.g., the maximum and minimum degrees in the graph) and (the convergent rate of) learning algorithms, lacking in-depth understanding of this relationship. It remains an open question how different characteristics of graph structure (e.g., power-law vs. regular) influence the performance of learning algorithms in the generalization regime (with noise). We aim to fill this gap by providing a framework for linking graph structure to the performance of learning algorithms in this setting.

**Excessive Risk of Learning Algorithm.** The excessive risk of different learning algorithms is a central research subject in the learning theory literature (Zou et al., 2021b; 2023; Tsigler & Bartlett, 2020; 2023; Dhillon et al., 2013; Lakshminarayanan & Szepesvari, 2018; Jain et al., 2017; Défossez & Bach, 2015). In particular, a key research question is how different learning algorithms perform under various settings (Dhillon et al., 2013). It remains unclear how the excessive risk of learning algorithms could manifest under graph learning and how different algorithms would perform in this setting. We expand the knowledge of the learning theory by providing an excessive risk analysis of SGD and Ridge regression in GNNs and comparing their

performance with respect to different graph structures.

**Spectral Graph Theory.** Spectral graph theory is a field of mathematics that studies the properties of graphs through the analysis of eigenvalues and eigenvectors of matrices associated with the graph, such as the adjacency matrix and the Laplacian matrix (Pósfai & Barabási, 2016; Spielman, 2012; Van Mieghem, 2023; Gera et al., 2018; Hammond et al., 2011; Chung, 1997). A foundation principle and result from the spectral graph theory is that the characteristic of the graph structure is closely coupled with the eigenspectrum of the graph matrix. For example, in power-law graphs, which are characterized by a few nodes with very high degrees and many nodes with low degrees, the eigenspectrum of the graph matrix is typically broad and heavy-tailed, reflecting the heterogeneity of the degree distribution (Faloutsos et al., 1999; Chung et al., 2003; Farkas et al., 2001; Goh et al., 2001; Easley et al., 2010). In contrast, regular graphs, where all nodes have the same degree, exhibit a more even eigenspectrum, reflecting the uniformity in the degree distribution (Easley et al., 2010). The comparison between power-law and regular graphs (visualized in Fig. 2 in Sec. 4) underscores the versatility of spectral graph theory in analyzing and interpreting the structural properties of different types of networks. These results serve as an important part of our toolkit for connecting the performance of the learning algorithm and graph structure.

## 3. Preliminaries

We introduce the notation, necessary background, and problem formulation in this paper.

**Notation.** We use lowercase letters to denote scalars and use lower and uppercase boldface letters to denote vectors and matrices. For a vector $\mathbf{x}$, $\mathbf{x}[i]$ denotes the $i$-th coordinate. For two functions $f(x) \geq 0$ and $g(x) \geq 0$ defined on $x > 0$, we write $f(x) \lesssim g(x)$ if $f(x) \leq c \cdot g(x)$ for some absolute constant $c > 0$; we write $f(x) \gtrsim g(x)$ if $g(x) \lesssim f(x)$; we write $f(x) \asymp g(x)$ if $f(x) \lesssim g(x) \lesssim f(x)$. For a vector $\boldsymbol{\theta} \in \mathbb{R}^d$ and a positive semidefinite matrix $\mathbf{H} \in \mathbb{R}^{d \times d}$, we denote $\|\boldsymbol{\theta}\|_{\mathbf{H}}^2 := \boldsymbol{\theta}^\top \mathbf{H} \boldsymbol{\theta}$. We denote $[k] := \{1, \ldots, k\}$.

**Graph.** A (standard) graph $\mathcal{G}$ with $N$ vertices is given by a two-tuple $(\mathcal{V}, \mathcal{E})$, where $\mathcal{V} = \{v_1, \ldots, v_N\}$ is the set of vertex and $\mathcal{E} \subset \mathcal{V} \times \mathcal{V}$ is a set of edge that defines the graph structure among $\mathcal{V}$. Each vertex $i \in [N]$ is associated with a feature vector $\mathbf{x}_i \in \mathcal{H}$ in some separable Hilbert space $\mathcal{H}$. The dimensionality of $\mathcal{H}$ is denoted as $d$, which can be infinite-dimensional when applicable. Let $\mathbf{X} \in \mathbb{R}^{N \times d}$ denote the feature matrix of all the vertices.

**Graph Neural Networks.** The computation in GNNs can be viewed as message-passing along graph structure (Jegelka, 2022). At each round $l$, the new embedding $\mathbf{x}_i^{(l)}$ for vertex $i$ is updated through a series of aggregate and combine steps as outlined below:

$$\mathbf{m}_i^{(l)} = \text{AGGREGATE}(\{\mathbf{x}_j^{(l-1)} \in \mathcal{N}(i)\}),$$
$$\mathbf{x}_i^{(l)} = \text{COMBINE}(\mathbf{x}_i^{(l-1)}, \mathbf{m}_i^{(l)}),$$

where $x_i^{(0)}$ is initialized as the feature vector $\mathbf{x}_i$, $\mathcal{N}(i)$ represents the neighbors of vertex $i$, and $\mathbf{m}_i$ denotes the aggregated representation at round $l$. Different GNNs differ in specific implementation of the AGGREGATE and COMBINE functions. Essentially, a GNN serves as an embedding function that integrates the graph structure and node features to produce an aggregated representation vector of node $v$. This representation is subsequently processed by a read-out function (e.g., a ReLU layer) to generate predictions.

**One-round Graph Neural Network.** In this study, we follow a similar setting to (Awasthi et al., 2021) and focus our main discussions on a one-round GNN which consists of an aggregation operation and a readout operation. Here we interpret the aggregation operation as a graph matrix $\mathbf{G}$ operating on the feature matrix $\mathbf{X}$. Then, we formulate the one-round GNNs with a two-component formulation. The first component is to aggregate the feature with the given aggregation operator and a graph matrix $\mathbf{G}$. We denote the resulting space as $\mathcal{M}$:

$$\mathcal{M} = \mathbf{G} \circ \mathbf{X}. \tag{3.1}$$

$\mathcal{M}$ can be viewed as the space where the aggregation representation lives. Without loss of generality, we assume $\mathcal{M}$ is a subspace of $\mathcal{H}$. Depending on the choice of graph matrix $\mathbf{G}$, we can recover different variants of GNNs (we provide a further discussion in this regard in Appendix G). For example, when $\mathbf{G} = \widehat{\mathbf{A}}$ (the normalized adjacency matrix of the graph) and $\circ$ is the simple matrix multiplication, we recover GCN (Kipf & Welling, 2017). Furthermore, for a given feature vector $\mathbf{x}$ and aggregation representation $\mathbf{m}$, we use

$$\mathbf{H} := \mathbb{E}_{\mathbf{x} \sim \mathbf{X}}[\mathbf{x}\mathbf{x}^\top], \quad \mathbf{M} := \mathbb{E}_{\mathbf{m} \sim \mathcal{M}}[\mathbf{m}\mathbf{m}^\top],$$

to denote the second moment of $\mathbf{m}$ and $\mathbf{x}$, which is the covariance matrix characterizing how the components of

$\mathbf{m}$ and $\mathbf{x}$ vary together. Then, we use $y \in \mathbb{R}$ to denote a response and it is generated with a ReLU readout:

$$y = \text{ReLU}(\mathbf{m}^\top \boldsymbol{\theta}^*) + \epsilon, \quad \mathbf{m} \sim \mathcal{M},$$

where $\boldsymbol{\theta}^* \in \mathcal{H}$ represents the unknown true model parameter and $\epsilon \in \mathbb{R}$ is the model noise with zero mean. It should be noted that there can be correlations among $\mathbf{m}$ manifested in the covariance of $\epsilon$. As a result, our generating model is very general. In addition, the above formulation is equivalent to the Markov-blanket formulation commonly used in other theoretical studies of GNNs (Wu et al., 2022).

**Learning Problem.** The goal of the learning problem is to estimate the true parameter $\boldsymbol{\theta}^*$ by minimizing the following:

$$L(\boldsymbol{\theta}^*) = \min_{\boldsymbol{\theta} \in \mathcal{H}} L(\boldsymbol{\theta}),$$
$$L(\boldsymbol{\theta}) := \frac{1}{2}\mathbb{E}\big[(y - \text{ReLU}(\mathbf{m}^\top \boldsymbol{\theta})^2\big],$$

where the expectation is taken over the randomness of $y$ and $\mathbf{m}$. The generalization performance of an estimated $\boldsymbol{\theta}$ found by a learning algorithm (e.g., SGD) is evaluated based on the excessive risk:

$$\Delta(\boldsymbol{\theta}) := L(\boldsymbol{\theta}) - L(\boldsymbol{\theta}^*).$$

The excessive risk $\Delta(\boldsymbol{\theta})$ can be decomposed into bias and variance as follows:

$$\Delta(\boldsymbol{\theta}) = \underbrace{\|\mathbb{E}[\boldsymbol{\theta}] - \boldsymbol{\theta}^*\|_{\mathbf{M}}^2}_{\text{bias}} + \underbrace{\|\boldsymbol{\theta} - \mathbb{E}[\boldsymbol{\theta}]\|_{\mathbf{M}}^2}_{\text{variance}}.$$

The decomposition above shows that the excess risk profile of $\Delta$ gives a comprehensive characterization of the learning algorithm, showcasing its convergence (governed by bias) and its robustness to noise (governed by variance). Following the prior literature (Bartlett et al., 2020; Zou et al., 2021b; Jain et al., 2017; Bach & Moulines, 2013; Berthier et al., 2020), we make the following assumptions on the data feature $\mathbf{H}$ and $\mathbf{G}$.

**Assumption 3.1.** $\mathbf{H}$ is positive definite (PD) and its trace $\text{tr}(\mathbf{H})$ is finite.

**Assumption 3.2.** There exists positive constant $\alpha$ such that for every $\mathbf{x}$ and any positive semidefinite matrix $\mathbf{A}$, it holds that: $\mathbb{E}[\mathbf{x}\mathbf{x}^\top \mathbf{A}\mathbf{x}\mathbf{x}^\top] - \mathbf{H}\mathbf{A}\mathbf{H} \preceq \alpha \, \text{tr}(\mathbf{A}\mathbf{H}) \cdot \mathbf{H}$.

**Assumption 3.3.** There exists a positive constant $\sigma$ such that: $\mathbb{E}_{\mathbf{x},\epsilon}[\epsilon^2 \mathbf{x}\mathbf{x}^\top] \leq \sigma^2 \mathbf{H}$.

**Assumption 3.4.** $\mathbf{G}$ is PD with respect to $\mathbf{x}$ and has a bounded norm.

This additional assumption on the graph matrix generally holds for common choices of graph matrices, such as the Laplacian and Adjacency matrices. To relate the graph

structure to the excessive risk of the learning algorithm, we are interested in the connection between $\boldsymbol{\theta}^*$ and $\mathbf{M}$. To analyze their relation, we introduce the following notation:

$$\mathbf{M}_{0:k} := \sum_{i=1}^{k} \mu_i \mathbf{v}_i \mathbf{v}_i^\top, \quad \mathbf{M}_{k:\infty} := \sum_{i>k} \mu_i \mathbf{v}_i \mathbf{v}_i^\top,$$

where $\{\mu_i\}_{i=1}^{\infty}$ are the eigenvalues of $\mathbf{M}$ sorted in non-increasing order and $\mathbf{v}_i$'s are the corresponding eigenvectors. Then we define:

$$\|\boldsymbol{\theta}\|_{\mathbf{H}_{0:k}^{-1}}^2 = \sum_{i \leq k} \frac{(\mathbf{v}_i^\top \boldsymbol{\theta})^2}{\mu_i}, \quad \|\boldsymbol{\theta}\|_{\mathbf{H}_{k:\infty}}^2 = \sum_{i>k} \mu_i (\mathbf{v}_i^\top \boldsymbol{\theta})^2.$$

## 4. Main Results

We next present the main results of this paper, focusing on the description and implication of the results. Detailed proof of the results can be found in the supplementary material.

**Result Overview.** We first derive the excessive risk profile (upper bound and lower bound) of learning algorithms (SGD and Ridge) in GNNs. These excessive profiles provide a connection between the graph matrix and the excessive risk of learning algorithms in GNNs. Then we use spectral graph theory to further establish the connection between the performance of the learning algorithm and graph structure. In particular, we investigate how the connectivity structure of the graphs (power-law vs. regular graph) affects the performance of different learning algorithms in GNNs. Furthermore, we extend the analysis to multi-layer linearized GNNs and yield a novel perspective on the over-smoothing issue of GNNs. This analysis reveals that the over-smoothing issue may be attributed to increasing excessive risk in learning algorithms due to a misalignment between the ground truth and the eigenspectrum of the aggregation space. Interestingly, increasing the number of GNN layers can be advantageous when the ground truth aligns with the headspace of the aggregation space. A visualization of the overall analytical framework is provided in Fig. 1.

### 4.1. Excessive Risk Profile of Learning Algorithms

We derive the excess risk profile that expands the current knowledge of learning theory and GNN, and forms a foundation for our further investigations on the effect of graph structure.

**SGD.** We consider SGD with a constant step size and tail-averaging (Bach & Moulines, 2013; Jain et al., 2017; 2018). At the $t$-th iteration, a fresh example is sampled from the aggregation space ($\mathbf{m}_t \sim \mathbf{G} \circ \mathbf{X}$) to perform the update:

$$\boldsymbol{\theta}_{t+1} = \boldsymbol{\theta}_t - \gamma \cdot \nabla l(\boldsymbol{\theta}_t; \mathbf{m}_t, y_t),$$
$$= \boldsymbol{\theta}_t - \gamma \cdot \left(\mathrm{ReLU}(\mathbf{m}_t^\top \boldsymbol{\theta}_{t-1}) - y_t\right) \cdot \mathbf{m}_t,$$

where $\gamma > 0$ is a constant stepsize (also referred to as the learning rate). After $N$ iterations, which corresponds to the number of samples observed, SGD computes the final estimator as the tail-averaged iterates:

$$\boldsymbol{\theta}_{\mathrm{sgd}}(N, \mathbf{G}; \gamma) := \frac{2}{N} \sum_{t=N/2}^{N-1} \boldsymbol{\theta}_t.$$

Here we focus on tail-average SGD [1], as it has been shown to improve the convergence properties and robustness of last-iterate SGD, particularly in the presence of noise and heavy-tailed data distributions (Zou et al., 2021b). The following theorem gives a description of the excessive risk profile of SGD under our setting.

**Theorem 4.1** (Excessive Risk Profile of SGD). *Consider SGD with tail-averaging with initialization $\boldsymbol{\theta}_0 = \mathbf{0}$. Suppose Assumptions 3.1, 3.2, 3.3 and 3.4 hold and stepsize satisfies $\gamma \leq 1/\operatorname{tr}(\mathbf{M})$. Then the excessive risk of SGD can be upper-bounded as follows:*

$$\Delta(\boldsymbol{\theta}_{\mathrm{sgd}}(N, \mathbf{G}; \gamma)) \lesssim \mathrm{SGDBias} + \mathrm{SGDVariance},$$

$$\mathrm{SGDBias} =$$
$$\frac{1}{\gamma^2 N^2} \left\| \exp\left(-N\gamma\mathbf{M}\right)\boldsymbol{\theta}^* \right\|_{\mathbf{M}_{0:k_1}^{-1}}^2 + \left\|\boldsymbol{\theta}^*\right\|_{\mathbf{M}_{k_1:\infty}}^2,$$
$$\mathrm{SGDVariance} =$$
$$\frac{\sigma^2 + \|\boldsymbol{\theta}^*\|_{\mathbf{M}}^2}{N} \cdot \left(k_2 + N^2\gamma^2 \sum_{i>k_2} \mu_i^2\right),$$

*where $k_1, k_2 \in [d]$ are arbitrary.*

*Suppose the stepsize satisfies $\gamma \leq 1/\mu_1$. Then the excess risk can be lower-bounded as follows:*

$$\Delta(\boldsymbol{\theta}_{\mathrm{sgd}}(N, \mathbf{G}; \gamma)) \gtrsim \mathrm{SGDBias} + \mathrm{SGDVariance},$$

$$\mathrm{SGDBias} =$$
$$\frac{1}{\gamma^2 N^2} \left\| \exp\left(-N\gamma\mathbf{M}\right)\boldsymbol{\theta}^* \right\|_{\mathbf{M}_{0:k^*}^{-1}}^2 + \left\|\boldsymbol{\theta}^*\right\|_{\mathbf{M}_{k^*:\infty}}^2,$$
$$\mathrm{SGDVariance} =$$
$$\frac{\sigma^2}{N} \cdot \left(k^* + N^2\gamma^2 \sum_{i>k^*} \mu_i^2\right) + \|\boldsymbol{\theta}^*\|_{\mathbf{M}}^2 \frac{\gamma}{\mu_1} \sum_{i>k^\dagger} \mu_i^2,$$

*where $k^* = \max\{k : \mu_k \geq 1/(N\gamma)\}$, and $k^\dagger = \max\{k : \mu_k \geq 2/(3N\gamma)\}$.*

Theorem 4.1 establishes the excessive risk upper and lower bounds for SGD in GNNs, offering a detailed characterization of each component (bias and variance) that constitutes the excessive risk profile of SGD. Several observations are

---

[1] we provide a more in-depth discussion of tail-average SGD in the supplementary material.

pertinent here. First, Theorem 4.1 demonstrates that the excessive risk of SGD (for both upper and lower bounds) is intimately linked to the aggregation matrix $\mathbf{M}$, which is derived from the graph matrix $\mathbf{G}$. This establishes a clear relationship between the performance of SGD and the graph matrix. Second, Theorem 4.1 reveals that the headspace and tail space of the eigenspectrum of $\mathbf{M}$ differently affect both the bias and variance. Specifically, the leading eigenspectrum shows an exponential decay in bias, indicating that SGD can effectively and efficiently approximate the ground truth associated with these directions. Third, while the decomposition of the eigenspectrum in the upper bound of SGD applies for any arbitrary $k_1, k_2 \in [d]$, the decomposition in the lower bound of SGD is more stringent and depends on the sample complexity and learning rate.

**Ridge Regression.** Given graph matrix $\mathbf{G}$ and feature matrix $\mathbf{X}$, Ridge regression provides an estimator for the true parameter by solving the following optimization problem:

$$\boldsymbol{\theta}_{\mathrm{ridge}}(N, \mathbf{G}; \lambda) := \arg\min_{\boldsymbol{\theta} \in \mathcal{H}} \|(\mathbf{G} \circ \mathbf{X})^\top \boldsymbol{\theta} - \mathbf{y}\|_2^2 + \lambda \|\boldsymbol{\theta}\|^2,$$

where $\lambda \geq 0$ is the regularization parameter. When $\lambda = 0$, the Ridge estimator reduces to the ordinary least square estimator (Hastie et al., 2009). In the following theorem, we give a characterization of the excessive risk profile of Ridge in the proposed setting.

**Theorem 4.2** (Excessive Risk Profile for Ridge). *Consider ridge regression with parameter $\lambda > 0$. Suppose Assumptions 3.1, 3.2, 3.3 and 3.4 hold. For a constant $\widehat{\lambda}$ depending on $\lambda$ and $N$, Ridge has the following excessive risk upper bound for an arbitrary $k \in [d]$:*

$$\Delta(\boldsymbol{\theta}_{\mathrm{ridge}}(N, \mathbf{G}; \lambda)) \lesssim \underbrace{\frac{\widehat{\lambda}^2}{N^2} \cdot \|\widehat{\boldsymbol{\theta}}^*\|_{\mathbf{M}_{0:k}^{-1}}^2 + \|\widehat{\boldsymbol{\theta}}^*\|_{\mathbf{M}_{k:\infty}}^2}_{\mathrm{RidgeBias}}$$

$$+ \underbrace{\sigma^2 \cdot \left( \frac{k}{N} + \frac{N}{\widehat{\lambda}^2} \sum_{i > k} \mu_i^2 \right)}_{\mathrm{RidgeVariance}}.$$

*Suppose $k^* = \min\{k : N\mu_k \lesssim \widehat{\lambda}\}$. Then we have the following excessive risk lower bound for Ridge:*

$$\Delta(\boldsymbol{\theta}_{\mathrm{ridge}}(N, \mathbf{G}; \lambda)) \gtrsim \underbrace{\frac{\widehat{\lambda}^2}{N^2} \cdot \|\widehat{\boldsymbol{\theta}}^*\|_{\mathbf{M}_{0:k^*}^{-1}}^2 + \|\widehat{\boldsymbol{\theta}}^*\|_{\mathbf{M}_{k^*:\infty}}^2}_{\mathrm{RidgeBias}}$$

$$+ \underbrace{\sigma^2 \cdot \left( \frac{k^*}{N} + \frac{N}{\widehat{\lambda}^2} \sum_{i > k^*} \mu_i^2 \right)}_{\mathrm{RidgeVariance}}.$$

Theorem 4.2 establishes an excessive risk profile (upper and lower bound) for Ridge regression in GNNs. Mirroring

Theorem 4.1, Theorem 4.2 demonstrates that the excessive risk of GNNs can be divided into two components, bias and variance, highlighting a structural similarity that is crucial for subsequent comparisons between the two learning algorithms. It is noted that the upper bound and lower bound of the excessive risk for Ridge generally aligns better than those for SGD, consistent with existing studies (Tsigler & Bartlett, 2020; 2023). By selecting $k$ equal to $k^*$, we achieve a unified bound for the risk associated with Ridge, differing only by a constant factor.

### 4.2. Graph Structure and Learning Performance

Next, we deepen our understanding of the connection between graph structure and the excess risk of learning algorithms using spectral graph theory. Spectral graph theory facilitates the linkage of graph structure to the eigenspectrum of the graph matrix. Specifically, we focus our analysis on the power-law graph, which is characterized by an exponential decay in the degree sequence, and the regular graph, where each vertex has a uniform degree.

**Power-law vs. Regular Graph.** As discussed in Sec. 2, the graph matrix of a power-law graph typically exhibits concentrated eigenvalues (i.e., a large eigenvalue associated with a few eigenvectors), resulting in a fast-decay eigenspectrum. In contrast, the graph matrix of regular graphs typically shows a uniform distribution of eigenvalues (i.e., eigenvalues of roughly the same magnitude across all eigenvectors), which leads to a slow-decay eigenspectrum. A visualization of these characteristics is given in Fig. 2.

For precise analysis, we consider graphs with eigenspectrum from the following model:

$$\mu_i(\mathbf{G}) = 1/i^\beta, \tag{4.1}$$

where $\mu_i(\mathbf{G})$ represents the $i$-th eigenvalue of the graph matrix $\mathbf{G}$ and $\beta > 0$ is the constant that controls the rate of decay in the eigenvalues. A larger $\beta$ results in a faster decay of the eigenspectrum, similar to that seen in power-law graphs, while a smaller $\beta$ leads to a more uniform eigenspectrum, akin to that of regular graphs. Additionally, we assume that the underlying ground truth model parameters and the feature space align with the eigenspectrum of the graph matrix, reflecting typical scenarios in graph learning problems (Fortunato, 2010). Based on this model, we present results that compare the performance of SGD and Ridge in power-law and regular graphs, thereby characterizing how graph structure influences the performance of the learning algorithm.

**Theorem 4.3** (Effect of Graph Structure). *Suppose Assumptions 3.1, 3.2, 3.3 and 3.4 hold. Consider a power-law graph $\mathcal{G}_{\mathrm{p}}$ with graph matrix $\mathbf{G}_{\mathrm{p}}$ whose eigen-spectrum is characterized by Eq. 4.1 with a large enough $\beta$. Then for*

*every $\lambda$ for ridge regression, there exists a choice for $\gamma^*$ for SGD such that for sufficiently large $N$, we have*

$$\Delta(\boldsymbol{\theta}_{\mathrm{sgd}}(N, \mathbf{G}_{\mathrm{p}}; \gamma^*)) \lesssim \Delta(\boldsymbol{\theta}_{\mathrm{ridge}}(N, \mathbf{G}_{\mathrm{p}}; \lambda)).$$

*On the other hand, consider a regular graph $\mathcal{G}_{\mathrm{r}}$ with graph matrix $\mathbf{G}_{\mathrm{r}}$ whose eigen-spectrum is characterized by Eq. 4.1 with a small $\beta$. Then for every choice of $\gamma$ for SGD, there exists a $\lambda^*$ such that,*

$$\Delta(\boldsymbol{\theta}_{\mathrm{ridge}}(N, \mathbf{G}_{\mathrm{r}}; \lambda^*)) \lesssim \Delta(\boldsymbol{\theta}_{\mathrm{sgd}}(N, \mathbf{G}_{\mathrm{r}}; \gamma)).$$

Theorem 4.3 suggests that SGD generally performs better (exhibits lower excessive risk) than Ridge regression in power-law graphs, characterized by a faster-decay graph spectrum. Conversely, Ridge regression tends to outperform SGD in regular graphs, which are characterized by a slower-decay graph spectrum. This result affirmatively supports our research question and establishes a connection between graph structure and learning algorithms via the graph spectrum. It also provides crucial insights for practical learning algorithm selection: an SGD-like learning algorithm is preferred for power-law graphs, while a Ridge-like algorithm is advisable for regular graphs.

*Remark* 4.4. Our theoretical results are not tightly dependent on the exact form of the power-law decay model. We adopt this model primarily for analytical clarity, but the core insights extend to a broader class of spectral decay behaviors. The essence of our findings (e.g., Theorem 4.3) lies in the rate at which the eigenvalues of the graph matrix decay, regardless of whether this follows a strict power-law. A faster-decaying spectrum—as often observed in graphs with power-law-like structures—tends to favor SGD. In contrast, a slower-decaying spectrum—as seen in more regular graphs—makes Ridge more favorable.

### 4.3. Linear GNNs and Over-smoothing

We extend our previous analysis to multi-layer GNNs and examine the cascading effect of stacking multiple layers. To make the analysis tractable, we focus on linearized GNNs (with ReLU readout) such as SGCN (Wu et al., 2019), following the approach in (Awasthi et al., 2021). We analyze the following $L$-layer linear GNN model:

$$\mathbf{M}^{(l)} = \mathbf{G} \circ \mathbf{M}^{(l-1)}, \quad 1 \le l \le L,$$
$$y = \mathrm{ReLU}\left(\left(\mathbf{m}^{(L)}\right)^\top \boldsymbol{\theta}^*\right) + \epsilon, \quad \mathbf{m}^{(L)} \sim \mathbf{M}^{(L)},$$

with the same assumptions as previously established.

**A New Perspective on Over-smoothing.** Over-smoothing in GNNs describes the phenomenon that the performance of a GNN degrades as the number of layers increases (Rusch et al., 2023). Previous studies have explored this issue

through the lens of node representation, attributing over-smoothing to the homogenization of representations (where iterative aggregation of neighbor information causes all nodes to converge to a similar representation), thereby losing the unique structural and feature information that distinguishes them. With our established analysis and results, we present the following novel implications and connections between learning algorithms and over-smoothing.

By recursively expanding the expression of multi-layer GNNs as described, we can analyze the $L$-layer GNNs through the modified graph matrix

$$\widehat{\mathbf{G}}(L) = \prod_{i=1}^{L} \mathbf{G}.$$

**Proposition 4.5** (Effect of Stacking GNN Layers). $\widehat{\mathbf{G}}$ *shares the same eigenbasis with $\mathbf{G}$. Furthermore, for two eigenvalues $\mu_i, \mu_j$ and a positive integer $l$, if $\mu_i > \mu_j$, then we have*

$$\frac{\mu_i(\widehat{\mathbf{G}}(l+1))}{\mu_j(\widehat{\mathbf{G}}(l+1))} > \frac{\mu_i(\widehat{\mathbf{G}}(l))}{\mu_j(\widehat{\mathbf{G}}(l))}.$$

Proposition 4.5 indicates that stacking additional GNN layers amplifies the relative difference (ratio) between the eigenvalues of different eigen-directions, thereby imposing a non-isotropic effect on the eigenspectrum of $\mathbf{G}$. More specifically, this leads to an eigenspectrum that exhibits faster decay. Building on this, and using the previous excessive risk analysis, we can further connect the performance of learning algorithms to the increasing number of GNN layers.

Most real-life graphs, particularly those used for academic benchmarks, are power-law graphs. As previously discussed, power-law graphs exhibit concentrated eigenspectra and feature a fast-decay spectrum. This creates a significant relative difference between a few directions and the rest of the directions on the eigenbasis. Accordingly, stacking additional layers of GNNs further amplify this relative difference. Based on our previous excessive risk analysis, if the ground truth does not align with the eigenspectrum of the graph matrix, increasing the number of GNN layers exacerbate this misalignment, leading to greater excessive risk (making learning more difficult for the learning algorithm and causing over-smoothing). Conversely, if the ground truth is concentrated in the few large eigen-directions, then adding more layers to GNNs will improve this alignment and consequently lead to a better excessive risk.

### 4.4. Empirical Study

We present an empirical study to validate our theoretical results and to illustrate the following:

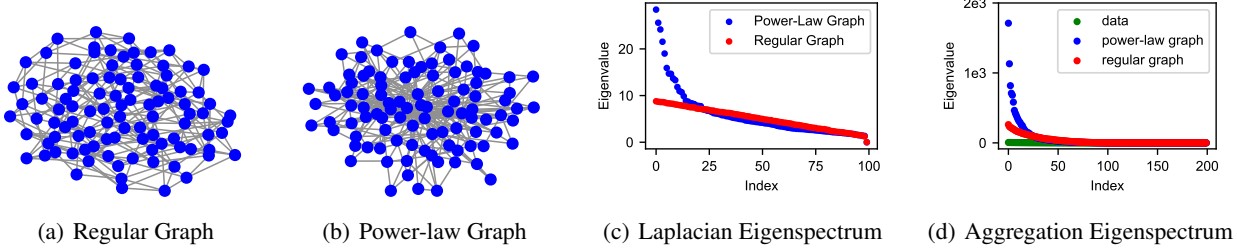

(a) Regular Graph     (b) Power-law Graph     (c) Laplacian Eigenspectrum     (d) Aggregation Eigenspectrum

*Figure 2.* Illustration of the Graph Structure and Eigenspectrum of Power-law and Regular Graphs. Fig. 2(a) and Fig. 2(b) illustrate regular and power-law graphs, respectively. Fig. 2(c) plots eigenspectrum of the graph Laplacian associated with the graphs. Fig. 2(d) shows eigenspectrum of the aggregation covariance space $\mathbf{M}$.

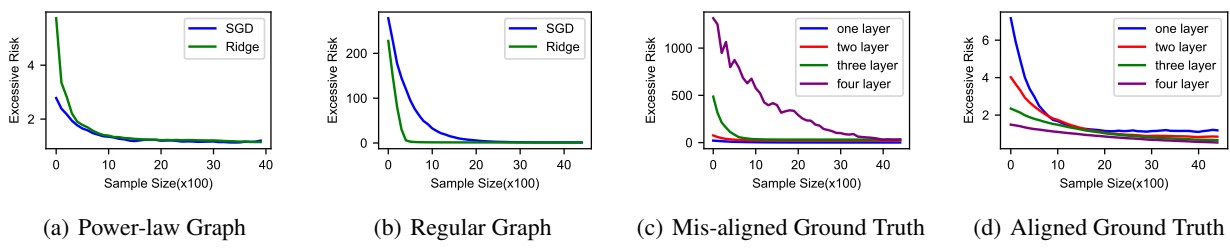

(a) Power-law Graph     (b) Regular Graph     (c) Mis-aligned Ground Truth     (d) Aligned Ground Truth

*Figure 3.* Illustration of the Performance of Learning algorithm in Different Graph Structures and the Effect of Stacking GNN Layers. Fig. 3(a) and 3(b) show performance comparison of SGD and Ridge in power-law and regular graph. Fig. 3(c) and 3(d) show the effect of stacking GNN layers with/without the ground truth concentrating in the head eigenspace.

1. the relation between the eigenspectrum of mixing space and graph structure (power-law vs. regular).

2. validate our theoretical prediction of SGD & Ridge in power-law and regular graphs.

3. the cascading effect when stacking multi-layer GNNs.

### 4.5. Setting

To control variations in graph structures, features, and model parameters, we focus our empirical study on graph simulation models, similar to other learning theory studies (Awasthi et al., 2021). In particular, we utilize the well-known Barabasi-Albert model (Pósfai & Barabási, 2016) to generate power-law graphs. For each node in the graph, we generate a feature vector of dimension 200 from the standard Gaussian distribution $\mathbb{N}(0, \mathbf{I})$. We generate the ground truth $\boldsymbol{\theta}^*$ that aligns with the eigenspectrum of the graph matrix. The model noise variance is set as $\sigma^2 = 1$. Each experiment is conducted for five independent trials, and the average results are reported.

### 4.6. Result

The empirical results are summarized in Fig. 2 and 3. Fig. 2(d) shows that the eigenspectrum of the aggregation space $\mathbf{M}$ aligns with the eigenspectrum of the graph structure, indicating that the eigenspectrum of the aggregation

space (induced by power-law vs. regular graphs) mirrors the eigenspectrum of the corresponding graph matrix (power-law vs. regular). Fig. 3(a) and 3(b) offer an end-to-end comparison of the performance of SGD and Ridge regression in power-law and regular graphs, respectively. These figures demonstrate that SGD outperforms Ridge regression in power-law graphs, while Ridge regression excels over SGD in regular graphs, validating our theoretical predictions. Fig. 3(c) illustrates that when the ground truth does not align well with the eigenspectrum (with ground truth vectors distributed in the tail eigenspace where $\mu_i$ is small), stacking GNN layers results in increased excessive risk, thereby causing the over-smoothing issue. Conversely, Fig. 3(d) indicates that increasing the number of layers can be beneficial when the ground truth aligns well (with ground truth concentrated in the head eigenspace where $\mu_i$ is large).

## 5. Conclusion and Discussion

### 5.1. Conclusion

In this paper, we conduct a theoretical investigation into the interplay between graph structure and learning algorithms in GNNs. Specifically, we extend the excessive risk analysis to include two core learning algorithms (SGD and Ridge) within the context of GNNs. We further utilize spectral graph theory to link the excessive risk of learning algo-

rithms with graph structure and carry out a comparative study across different graph structures (power-law and regular). Our findings indicate that SGD generally outperforms Ridge in power-law graphs and the reverse holds true in regular graphs. This positively answers our research question regarding whether graph structure can influence the performance of learning algorithms and demonstrates the structural relationship between graph structure and learning algorithms.

## 5.2. Future Work

This study opens up intriguing directions for future research. First, our analysis was confined to one-layer GNNs or linearized GNNs to maintain traceability in the theoretical analysis. This limitation is common in similar studies, even with I.I.D data, due to the constraints of existing theoretical tools. Developing more sophisticated analytical tools to extend the analysis to more complex models, such as those with multiple non-linear layers, would be beneficial. Second, given our focus on the interplay between graph structure and learning algorithms, we isolated the relationship and impact of graph structure and node features. Exploring how a structural relationship between graph structure and node features could further influence learning algorithms, would also be a worthwhile direction, particularly regarding its implications on over-squashing (Topping et al., 2021).

Finally, extending our theoretical framework to random graph models constitutes another promising avenue. While our current results can directly apply to specific realizations of random graphs, a comprehensive probabilistic treatment—considering expected or high-probability spectral behaviors—would generalize our findings significantly. Leveraging concentration inequalities or advanced random matrix theory to derive probabilistic guarantees on the performance differences between SGD and Ridge regression across graph ensembles (e.g., Erdős–Rényi, stochastic block models, and other popular random graph families) represents a meaningful future research direction.

## Impact Statement

This paper presents work whose goal is to advance the understanding of graph neural networks. There are many potential societal consequences of our work, none which we feel must be specifically highlighted here.

## Acknowledgement

We would like to thank the anonymous reviewers and area chairs for their helpful comments. This work was supported in part by grants from Hong Kong RGC under the contracts 17207621, 17203522, and C7004-22G (CRF).

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

# A. Excessive Risk of SGD

In this appendix, we provide proof of the excessive risk (Theorem 4.1) of SGD in our proposed setting. In this part, we will mainly follow the proof technique and results in (Zou et al., 2021b) that is developed to sharply characterize the excess risk bound for SGD (with tail-averaging) when the data distribution has a nice finite fourth-moment bound. However, such a condition does not directly apply to the case with aggregation and ReLU activation. Therefore, their result can not be directly applied here.

## A.1. Preliminary and Useful Lemmas

We start with stating some commonly used notation and proving a set of lemmas that are going to be useful for the subsequent analysis. First, let us recall some common notation and definitions that are going to be used in the proof. We denote the node feature matrix $\mathbf{X}$ and $d$, $\mathbf{G}$ to be the graph matrix associated with the GNN, and $\mathbf{M}$ to be the aggregation space after applying the graph matrix $\mathbf{G}$ on $\mathbf{X}$. Furthermore, we denote

$$\mathbf{H} = \mathbb{E}[\mathbf{x}\mathbf{x}^\top], \quad \mathbf{M} = \mathbb{E}[\mathbf{m}\mathbf{m}^\top],$$

to be the covariance of data distribution and aggregation distribution. We denote

$$\mu_1, ..., \mu_d,$$

to be the eigenvalues for $\mathbf{M}$ in descending orders.

We start with proving the implication of the Assumptions 3.1, 3.3, and 3.2 on the aggregation space.

**Lemma A.1.** *Suppose Assumption 3.2 holds, there exists positive constant $\alpha'$ such that for every $\mathbf{m}$ and any positive semidefinite matrix $\mathbf{A}$, it holds that:* $\mathbb{E}[\mathbf{m}\mathbf{m}^\top\mathbf{A}\mathbf{m}\mathbf{m}^\top] - \mathbf{M}\mathbf{A}\mathbf{M} \preceq \alpha' \operatorname{tr}(\mathbf{A}\mathbf{M}) \cdot \mathbf{M}$.

*Proof.* By definition, we have that

$$\mathbf{m} \sim \mathcal{M} := \mathbf{G} \circ \mathbf{X}.$$

For a given PSD matrix $\mathbf{A}$, we denote

$$\mathbf{B} = \mathbf{G}\mathbf{A}\mathbf{G}.$$

By Assumption 3.4, we have that $\mathbf{G}$ is a PSD matrix. Hence, we can immediately conclude that $\mathbf{B}$ is also a PSD matrix. Then, we have that,

$$\mathbb{E}[\mathbf{m}\mathbf{m}^\top\mathbf{A}\mathbf{m}\mathbf{m}^\top] - \mathbf{M}\mathbf{A}\mathbf{M},$$
$$= \mathbf{G}\mathbb{E}[\mathbf{x}\mathbf{x}^\top\mathbf{B}\mathbf{x}\mathbf{x}^\top]\mathbf{G} - \mathbf{G}\mathbf{X}\mathbf{X}^\top\mathbf{B}\mathbf{X}\mathbf{X}^\top\mathbf{G},$$
$$= \mathbf{G}\mathbb{E}[\mathbf{x}\mathbf{x}^\top\mathbf{B}\mathbf{x}\mathbf{x}^\top]\mathbf{G} - \mathbf{G}\mathbf{H}\mathbf{B}\mathbf{H}\mathbf{G},$$
$$= \mathbf{G}\left(\mathbb{E}[\mathbf{x}\mathbf{x}^\top\mathbf{B}\mathbf{x}\mathbf{x}^\top] - \mathbf{H}\mathbf{B}\mathbf{H}\right)\mathbf{G}.$$

Then, by Assumption 3.2, we have that there exists a $\alpha$ such that

$$\mathbb{E}[\mathbf{x}\mathbf{x}^\top\mathbf{B}\mathbf{x}\mathbf{x}^\top] - \mathbf{H}\mathbf{B}\mathbf{H} \preceq \alpha \operatorname{tr}(\mathbf{B}\mathbf{H}) \cdot \mathbf{H}$$

Substitute this back into the derivation above, we have that,

$$\mathbb{E}[\mathbf{m}\mathbf{m}^\top\mathbf{A}\mathbf{m}\mathbf{m}^\top] - \mathbf{M}\mathbf{A}\mathbf{M},$$
$$= \mathbf{G}\left(\mathbb{E}[\mathbf{x}\mathbf{x}^\top\mathbf{B}\mathbf{x}\mathbf{x}^\top] - \mathbf{H}\mathbf{B}\mathbf{H}\right)\mathbf{G},$$
$$\preceq \alpha \operatorname{tr}(\mathbf{B}\mathbf{G}\mathbf{H}\mathbf{G}) \cdot \mathbf{G}\mathbf{H}\mathbf{G}.$$

Again, by Assumption 3.4, we have that $\mathbf{G}$ is a bounded matrix and immediately obtain that there exists a $\alpha'$ such that,

$$\mathbb{E}[\mathbf{m}\mathbf{m}^\top\mathbf{A}\mathbf{m}\mathbf{m}^\top] - \mathbf{M}\mathbf{A}\mathbf{M},$$
$$\preceq \alpha \operatorname{tr}(\mathbf{G}\mathbf{A}\mathbf{G}\mathbf{G}\mathbf{H}\mathbf{G}) \cdot \mathbf{G}\mathbf{H}\mathbf{G},$$
$$= \alpha \operatorname{tr}(\mathbf{G})^2 \operatorname{tr}(\mathbf{A}\mathbf{G}\mathbf{H}\mathbf{G}) \cdot \mathbf{G}\mathbf{H}\mathbf{G},$$
$$= \alpha' \operatorname{tr}(\mathbf{A}\mathbf{M}) \cdot \mathbf{M}$$

□

**Lemma A.2.** *There exists a positive constant $\sigma'$ such that: $\mathbb{E}_{\mathbf{m},\epsilon}[\epsilon^2 \mathbf{m}\mathbf{m}^\top] \leq \sigma^2 \mathbf{M}$.*

*Proof.*

$$\mathbb{E}[\epsilon^2 \mathbf{m}\mathbf{m}^\top] = \mathbf{G}\mathbb{E}[\epsilon^2 \mathbf{x}\mathbf{x}^\top]\mathbf{G},$$

By Assumption 3.3, we have that,

$$\leq \mathbf{G}(\sigma^2 \mathbf{H})\mathbf{G},$$
$$= \sigma^2 \mathbf{G}\mathbf{H}\mathbf{G},$$
$$= \sigma^2 \mathbf{M}$$

□

**Lemma A.3.** *The covariance matrix of aggregation space $\mathbf{M}$ is PSD and has a finite trace.*

*Proof.* First recall that

$$\mathbf{M} = \mathbf{G} \circ \mathbf{X}.$$

By Assumption 3.1, we have that $\mathbf{X}$ is SD. Similarly, by Assumption 3.4, we have that $\mathbf{G}$ is SD. We immediately obtain that $\mathbf{M}$ is PSD.

Now, it remains to show that $\mathbf{M}$ has a finite trace.

$$\mathrm{tr}(\mathbf{M}) \lesssim \mathrm{tr}(\mathbf{X})\,\mathrm{tr}(\mathbf{G})$$

Again, by Assumption 3.1 and Assumption 3.4, we have that $\mathrm{tr}(\mathbf{X})$ and $\mathrm{tr}(\mathbf{G})$ are finite and therefore, $\mathrm{tr}(\mathbf{M})$ is finite. □

The analysis in (Bartlett et al., 2020; Zou et al., 2021b) established a sharp analysis for SGD and Ridge with bounded fourth-moment assumption. However, the setting considered in these two studies assumes a linear model without non-linear activation. In other words, the SGD update is given by,

$$\boldsymbol{\theta}_{t+1} = \boldsymbol{\theta}_t - \gamma \cdot (\mathbf{m}_t^\top \boldsymbol{\theta}_{t-1} - y_t) \cdot \mathbf{m}_t,$$

In our study, we consider GNN layers with ReLU activation and the update is given by,

$$\boldsymbol{\theta}_{t+1} = \boldsymbol{\theta}_t - \gamma \cdot (\mathrm{ReLU}(\mathbf{m}_t^\top \boldsymbol{\theta}_{t-1}) - y_t) \cdot \mathbf{m}_t,$$

We adopt the following result from (Wu et al., 2023) to relate the excessive risk landscape of SGD and SGD with ReLU activation.

**Lemma A.4** (Restatement of Lemma 4.2 in (Wu et al., 2023)). *The excessive risk landscape of ReLU updates is given by,*

$$0.25\|\boldsymbol{\theta} - \boldsymbol{\theta}^*\|_{\mathbf{M}}^2 \leq \Delta(\boldsymbol{\theta}) \leq \|\boldsymbol{\theta} - \boldsymbol{\theta}^*\|_{\mathbf{M}}^2$$

Lemma A.4 suggest that even though the excess risk induced by ReLU activation could be non-convex locally, the landscape of the excess risk on a large scale is "approximately" quadratic in the sense of ignoring some multiplicative factors. This landscape enables us to build sharp upper and lower bounds on the excess risk by bounding a simpler quadratic,

$$\|\boldsymbol{\theta} - \boldsymbol{\theta}^*\|_{\mathbf{M}}^2.$$

## A.2. SGD Excessive Risk

Next, we present proof for the excessive risk profile, Theorem 4.1, for SGD under GNN. For this, we decompose the proof and the results of the theorem into two parts: upper bound and lower bound that involve different results.

The overall proof structures are as follows: 1) we first derive the learning dynamic with the aggregation space under the linear model and 2) we relate the excessive risk landscape of the above learning dynamic with ReLU activation with the lemma proved above.

Next, we start with considering a simplified linear model where the response is generated by the aggregation space without the ReLU activation, which amount to the following generation model,

$$y = \mathbf{m}^\top \boldsymbol{\theta}^* + \epsilon, \quad \mathbf{m} \sim \mathcal{M}, \tag{A.1}$$

Let $\widehat{\boldsymbol{\theta}}_t$ be the t-iteration of the SGD under the generation model Eq. A.1 which amounts to the following process,

$$\begin{aligned}
\widehat{\boldsymbol{\theta}}_{t+1} &= \widehat{\boldsymbol{\theta}}_t - \gamma \cdot \nabla l(\widehat{\boldsymbol{\theta}}_t; \mathbf{m}_t, y_t), \\
&= \widehat{\boldsymbol{\theta}}_t - \gamma \cdot (\mathbf{m}_t^\top \widehat{\boldsymbol{\theta}}_{t-1} - y_t) \cdot \mathbf{m}_t.
\end{aligned} \tag{A.2}$$

Then, we denote

$$\boldsymbol{\eta}_t := \widehat{\boldsymbol{\theta}}_t - \boldsymbol{\theta}^*,$$

as the centered SGD iterate, and,

$$\bar{\boldsymbol{\eta}}_N = \frac{1}{N} \sum_{t=1}^N \boldsymbol{\eta}_t$$

Then, we want to decompose the SGD iterate into bias and variance, denoted as $\boldsymbol{\eta}^{\text{bias}}$ and $\boldsymbol{\eta}^{\text{var}}$ respectively. The corresponding update rules are given by,

$$\begin{aligned}
\boldsymbol{\eta}_t^{\text{bias}} &= (\mathbf{I} - \gamma \mathbf{m}_t \mathbf{m}_t^\top) \boldsymbol{\eta}_{t-1}^{\text{bias}}, \\
\boldsymbol{\eta}_0^{\text{bias}} &= \boldsymbol{\eta}_0^{\text{bias}}, \\
\boldsymbol{\eta}_t^{\text{variance}} &= (\mathbf{I} - \gamma \mathbf{m}_t \mathbf{m}_t^\top) \boldsymbol{\eta}_{t-1}^{\text{variance}} + \gamma \epsilon \mathbf{m}_t, \\
\boldsymbol{\eta}_0^{\text{variance}} &= \boldsymbol{\eta}_0^{\text{variance}},
\end{aligned}$$

Thanks to (Jain et al., 2017), we have the following results for the excessive risk decomposition

**Lemma A.5.** *Consider SGD iterates with a linear model. Then, the excessive risk of the SGD under a linear model can be decomposed into bias and variance as follows,*

$$\Delta(\widehat{\boldsymbol{\theta}}) = \frac{1}{2} \langle \mathbf{M}, \mathbb{E}[\bar{\boldsymbol{\eta}}_N \otimes \bar{\boldsymbol{\eta}}_N] \rangle \le (\sqrt{\text{bias}} + \sqrt{\text{variance}})^2, \tag{A.3}$$

*where*

$$\sqrt{\text{bias}} = \frac{1}{2} \langle \mathbf{M}, \mathbb{E}[\bar{\boldsymbol{\eta}}_N^{\text{bias}} \otimes \bar{\boldsymbol{\eta}}_N^{\text{bias}}] \rangle,$$

*and*

$$\sqrt{\text{variance}} = \frac{1}{2} \langle \mathbf{M}, \mathbb{E}[\bar{\boldsymbol{\eta}}_N^{\text{var}} \otimes \bar{\boldsymbol{\eta}}_N^{\text{var}}] \rangle,$$

*Proof.* The result can be obtained by simply following the procedure in (Jain et al., 2017) by treating $\mathbf{M}$ as the new data covariance matrix. $\square$

Then, thanks to the analysis in (Zou et al., 2021b), we have a comprehensive understanding of the above iterative process of SGD and have the following results.

**Lemma A.6.** *Under The bias and variance of the SGD iterates with respect to $\mathbf{M}$ is upper bounded by,*

$$\text{bias} := \frac{1}{2}\langle \mathbf{M}, \mathbb{E}[\bar{\boldsymbol{\eta}}_N^{\text{bias}} \otimes \bar{\boldsymbol{\eta}}_N^{\text{bias}}]\rangle,$$

$$\leq \frac{1}{N^2}\sum_{t=0}^{N-1}\sum_{k=t}^{N-1}\langle (\mathbf{I} - \gamma\mathbf{M})^{k-t}\mathbf{M}, \mathbb{E}[\bar{\boldsymbol{\eta}}_t^{\text{bias}} \otimes \bar{\boldsymbol{\eta}}_t^{\text{bias}}]\rangle,$$

*and*

$$\text{variance} := \frac{1}{2}\langle \mathbf{M}, \mathbb{E}[\bar{\boldsymbol{\eta}}_N^{\text{var}} \otimes \bar{\boldsymbol{\eta}}_N^{\text{var}}]\rangle,$$

$$\leq \frac{1}{N^2}\sum_{t=0}^{N-1}\sum_{k=t}^{N-1}\langle (\mathbf{I} - \gamma\mathbf{M})^{k-t}\mathbf{M}, \mathbb{E}[\bar{\boldsymbol{\eta}}_t^{\text{var}} \otimes \bar{\boldsymbol{\eta}}_t^{\text{var}}]\rangle.$$

*Proof.* The result can be obtained by simply following the proof for Lemma B.3 in (Zou et al., 2021b) by treating $\mathbf{M}$ as the new data covariance matrix. $\square$

Next, we present the proof for the upper bound and lower bound for the excessive risk of SGD.

**Lemma A.7** (SGD Excessive Risk Upper Bound). *Consider SGD with tail-averaging with initialization $\boldsymbol{\theta}_0 = \mathbf{0}$. Suppose Assumptions 3.1, 3.2, 3.3 and 3.4 hold and stepsize satisfies $\gamma \leq 1/\operatorname{tr}(\mathbf{M})$. Then the excessive risk of SGD under one-layer GNN can be upper-bounded as follows:*

$$\Delta(\boldsymbol{\theta}_{\text{sgd}}(N, \mathbf{G}; \gamma)) \lesssim \text{SGDBias} + \text{SGDVariance},$$

$$\text{SGDBias} = \frac{1}{\gamma^2 N^2}\big\| \exp\big(-N\gamma\mathbf{M}\big)\boldsymbol{\theta}^*\big\|_{\mathbf{M}_{0:k_1}^{-1}}^2 + \big\|\boldsymbol{\theta}^*\big\|_{\mathbf{M}_{k_1:\infty}}^2,$$

$$\text{SGDVariance} = \frac{\sigma^2 + \|\boldsymbol{\theta}^*\|_{\mathbf{M}}^2}{N}\cdot\bigg(k_2 + N^2\gamma^2\sum_{i>k_2}\mu_i^2\bigg),$$

*where $k_1, k_2 \in [d]$ are arbitrary.*

*Proof.* Consider the SGD iterates without ReLU activation that is given by,

$$\widehat{\boldsymbol{\theta}}_{t+1} = \widehat{\boldsymbol{\theta}}_t - \gamma\cdot(\mathbf{m}_t^\top\widehat{\boldsymbol{\theta}}_{t-1} - y_t)\cdot\mathbf{m}_t, \quad \mathbf{m}_t \sim \mathbf{M}.$$

The SGD iterates above amounts to linear model with respect to the aggregation space $\mathbf{M}$. Then, by Lemma A.1, Lemma A.3 and Lemma A.2, we get that the above SGD iterates with respect to $\mathbf{M}$ also has bounded fourth-moment and model noise.

This means that we can immediately apply the results from By Lemma A.4, we can derive the excessive risk by considering the simplified iterates without the ReLU activation that is given by Eq. A.2.

By Lemma A.1, and Lemma A.2, we have that the fouth-moment and the modelling noise of the SGD iterate with respect to the aggregation space is bounded. This means that we can immediately apply the bias variance decomposition and the SGD iterate bound on process above with respect to $\mathbf{M}$.

Then, we can follow the arguments in (Zou et al., 2021b) and obtain the desired result by involving Theorem 5.1 in (Zou et al., 2021b).

The ReLU activation does not affect the overall excessive risk landscape as given by Lemma A.4.

$\square$

**Lemma A.8** (SGD Excessive Risk Lower Bound). *Suppose the stepsize satisfies $\gamma \leq 1/\mu_1$. Then the excess risk can be lower-bounded as follows:*

$$\Delta(\boldsymbol{\theta}_{\text{sgd}}(N, \mathbf{G}; \gamma)) \gtrsim \text{SGDBias} + \text{SGDVariance},$$

$$\text{SGDBias} = \frac{1}{\gamma^2 N^2} \big\| \exp\big(-N\gamma \mathbf{M}\big)\boldsymbol{\theta}^* \big\|^2_{\mathbf{M}_{0:k^*}^{-1}} + \big\|\boldsymbol{\theta}^*\big\|^2_{\mathbf{M}_{k^*:\infty}},$$

$$\text{SGDVariance} =$$
$$\frac{\sigma^2}{N} \cdot \left( k^* + N^2 \gamma^2 \sum_{i>k^*} \mu_i^2 \right) + \|\boldsymbol{\theta}^*\|^2_{\mathbf{M}} \frac{\gamma}{\mu_1} \sum_{i>k^\dagger} \mu_i^2,$$

*where $k^* = \max\{k : \mu_k \geq 1/(N\gamma)\}$, and $k^\dagger = \max\{k : \mu_k \geq 2/(3N\gamma)\}$.*

*Proof.* The results are obtained through a similar argument as the upper but involving Theorem 5.2 □

Then, combining the two lemmas, upper bound and lower bound above, we immediately obtain the result for Theorem 4.1.

## B. Excessive Risk of Ridge

In this appendix, we present proof for characterizing the excessive risk of Ridge in our setting.

We denote $\mathbf{D} = \mathbf{G} \circ \mathbf{X}$ to be the data matrix associated with the aggregation space. Then, recall that the standard ridge regression is equivalent to the following least square problem,

$$\arg\min_{\boldsymbol{\theta}} \|\mathbf{D}\boldsymbol{\theta} - \mathbf{y}\|_2^2 + \lambda\|\boldsymbol{\theta}\|_2^2.$$

We denote $\boldsymbol{\theta}_{\text{Ridge}}(N; \lambda)$ to be the solution to the optimization problem above. Then, we have the following bias-variance decomposition of Ridge with respect to the aggregation space.

**Lemma B.1.** *For any $\lambda > 0$, we have that*

$$\Delta(\boldsymbol{\theta}_{\text{Ridge}}(N; \lambda)) = \text{RidgeBias} + \text{RidgeVariance},$$

*where*

$$\text{RidgeBias} =$$
$$\frac{\lambda}{2} \cdot \mathbb{E}\left[ (\boldsymbol{\theta}^*)^\top (\mathbf{D}^\top\mathbf{D} + \lambda\mathbf{I})^{-1}\mathbf{M}(\mathbf{D}^\top\mathbf{D} + \lambda\mathbf{I})^{-1}\boldsymbol{\theta}^* \right],$$

$$\text{RidgeVariance} =$$
$$\sigma^2 \cdot \mathbb{E}\left[ \text{tr}\left( (\mathbf{D}^\top\mathbf{D} + \lambda\mathbf{I})^{-1}\mathbf{D}^\top\mathbf{D}(\mathbf{D}^\top\mathbf{D} + \lambda\mathbf{I})^{-1}\mathbf{M} \right) \right],$$

*where the expectations are taken over the randomness of the training data matrix $\mathbf{D}$.*

**Proof.** First, it is known and easy to derive that the solution of ridge regression takes the form

$$\boldsymbol{\theta}_{\text{Ridge}}(N; \lambda) = (\mathbf{D}^\top\mathbf{D} + \lambda\mathbf{I})^{-1}\mathbf{D}^\top\mathbf{y},$$

where $\mathbf{D}$ is the data matrix and $\mathbf{y}$ is the response vector. Then, according to the definition of the loss function $L(\boldsymbol{\theta})$, we have

$$\mathbb{E}[L(\boldsymbol{\theta}_{\text{Ridge}}(N; \lambda))]$$
$$= \mathbb{E}\left[ \langle y - \langle \boldsymbol{\theta}_{\text{Ridge}}(N; \lambda), \mathbf{m} \rangle \rangle^2 \right],$$
$$= \mathbb{E}\left[ \langle \boldsymbol{\theta}^*, \mathbf{m} \rangle - \langle \boldsymbol{\theta}_{\text{Ridge}}(N; \lambda), \mathbf{m} \rangle \right]^2 + \mathbb{E}\left[ y - \langle \boldsymbol{\theta}^*, \mathbf{m} \rangle \right]^2$$
$$\quad + 2\mathbb{E}\left[ \langle \boldsymbol{\theta}^*, \mathbf{m} \rangle - \langle \boldsymbol{\theta}_{\text{Ridge}}(N; \lambda), \mathbf{m} \rangle \right] \cdot (y - \langle \boldsymbol{\theta}^*, \mathbf{m} \rangle)$$
$$= \mathbb{E}[\|\boldsymbol{\theta}_{\text{Ridge}}(N; \lambda) - \boldsymbol{\theta}^*\|_{\mathbf{M}}^2] + L(\boldsymbol{\theta}^*)$$

where the last equation follows from the modelling assumption that the expected value of noise is zero. Then, regarding $\mathbb{E}[\|\boldsymbol{\theta}_{\text{Ridge}}(N;\lambda) - \boldsymbol{\theta}^*\|_{\mathbf{M}}^2]$, let $\epsilon = y - \langle \boldsymbol{\theta}^*, \mathbf{m} \rangle$ be the model noise vector, we have

$$
\mathbb{E}[\|\boldsymbol{\theta}_{\text{Ridge}}(N;\lambda) - \boldsymbol{\theta}^*\|_{\mathbf{M}}^2]
$$
$$
= \mathbb{E}\left[\left\|(\mathbf{D}^\top\mathbf{D} + \lambda\mathbf{I})^{-1}\mathbf{D}\top\mathbf{y} - \boldsymbol{\theta}^*\right\|_{\mathbf{M}}^2\right],
$$
$$
= \mathbb{E}\left[\left\|(\mathbf{D}\top\mathbf{D} + \lambda\mathbf{I})^{-1}\mathbf{D}^\top(\mathbf{D}\boldsymbol{\theta}^* + \epsilon) - \boldsymbol{\theta}^*\right\|_{\mathbf{M}}^2\right],
$$
$$
= \underbrace{\mathbb{E}\left[\left\|(\mathbf{D}^\top\mathbf{D} + \lambda\mathbf{I})^{-1}\mathbf{D}^\top\mathbf{D}\boldsymbol{\theta}^* - \boldsymbol{\theta}^*\right\|_{\mathbf{M}}^2\right]}_{\text{bias}}
$$
$$
+ \underbrace{\mathbb{E}\left[\left\|(\mathbf{D}^\top\mathbf{D} + \lambda\mathbf{I})^{-1}\mathbf{D}^\top\epsilon\right\|_{\mathbf{M}}^2\right]}_{\text{variance}},
$$

where in the last inequality again follow from the modelling assumption that $\mathbb{E}[\epsilon|\mathbf{D}] = 0$. More specifically, the bias error can be reformulated as

$$
\text{RidgeBias} = \mathbb{E}\left[\left\|(\mathbf{D}^\top\mathbf{D} + \lambda\mathbf{I})^{-1}\mathbf{D}^\top\mathbf{D} - \mathbf{I}\right\|_{\mathbf{M}}^2 \boldsymbol{\theta}^*\right]
$$
$$
= \frac{\lambda}{2}\mathbb{E}\left[\left\|(\mathbf{D}^\top\mathbf{D} + \lambda\mathbf{I})^{-1}\boldsymbol{\theta}^*\right\|_{\mathbf{M}}^2\right]
$$
$$
= \frac{\lambda}{2}\mathbb{E}\left[(\boldsymbol{\theta}^*)^\top(\mathbf{D}^\top\mathbf{D} + \lambda\mathbf{I})^{-1}\mathbf{M}(\mathbf{D}^\top\mathbf{D} + \lambda\mathbf{I})^{-1}\boldsymbol{\theta}^*\right].
$$

In terms of the variance error, note that by Lemma A.2 we have $\mathbb{E}[\epsilon\epsilon^\top|\mathbf{D}] = \sigma^2\mathbf{I}$, then

$$
\text{RidgeVariance} = \mathbb{E}\left[\left\|(\mathbf{D}^\top\mathbf{D} + \lambda\mathbf{I})^{-1}\mathbf{D}^\top\epsilon\right\|_{\mathbf{M}}^2\right]
$$
$$
= \mathbb{E}\left[\text{tr}\left((\mathbf{D}^\top\mathbf{D} + \lambda\mathbf{I})^{-1}\mathbf{D}^\top\epsilon\epsilon^\top\mathbf{D}(\mathbf{D}^\top\mathbf{D} + \lambda\mathbf{I})^{-1}\mathbf{M}\right)\right]
$$
$$
= \sigma^2 \cdot \mathbb{E}\left[\text{tr}\left((\mathbf{D}^\top\mathbf{D} + \lambda\mathbf{I})^{-1}\mathbf{D}^\top\mathbf{D}(\mathbf{D}^\top\mathbf{D} + \lambda\mathbf{I})^{-1}\mathbf{M}\right)\right].
$$

With the bias-variance decomposition above, we can follow a simple extension of Lemmas 2 & 3 in (Tsigler & Bartlett, 2023) for characterizing the excessive risk of ridge regression. Next, we show how to extend their results into our setting. To do so, we decompose the result of Theorem 4.2 into upper bound and lower bound in a similar manner.

**Lemma B.2.** *Let $\lambda \geq 0$ be the regularization parameter, $n$ be the training sample size, $\mu_1, ..., \mu_d$ be the eigenvalues of $\mathbf{M}$ in descending order and $\boldsymbol{\theta}_{\text{ridge}}(N;\lambda)$ be the output of ridge regression. Then*

$$
\Delta(\boldsymbol{\theta}_{\text{ridge}}(N;\lambda)) = \text{RidgeBias} + \text{RidgeVariance},
$$

*and there is some absolute constant $b > 1$, such that for*

$$
k^* := \min\left\{k : b\mu_{k+1} \leq \frac{\lambda + \sum_{i>k}\mu_i}{n}\right\},
$$

*the following holds:*

$$
\text{RidgeBias} \gtrsim \left(\frac{\lambda + \sum_{i>k^*_{\text{ridge}}}\mu_i}{N}\right)^2 \cdot \|\boldsymbol{\theta}^*\|_{\mathbf{M}^{-1}_{0:k^*_{\text{ridge}}}}^2
$$
$$
+ \|\boldsymbol{\theta}^*\|_{\mathbf{M}_{k^*_{\text{ridge}}:\infty}}^2,
$$
$$
\text{RidgeVariance} \gtrsim \sigma^2 \cdot \left\{\frac{k^*_{\text{ridge}}}{N} + \frac{N\sum_{i>k^*_{\text{ridge}}}\mu_i^2}{\left(\lambda + \sum_{i>k^*_{\text{ridge}}}\mu_i\right)^2}\right\}.
$$

*Proof.* By involving Lemma A.4, we can relate the excessive risk of ReLU Ridge regression with the linear model. Then, by Lemma B.1 and applying Lemmas 2 & 3 in (Tsigler & Bartlett, 2023) for characterizing the excessive risk of ridge regression and the extension of Theorem B.2 in (Zou et al., 2021a) on the aggregation space $\mathbf{M}$, we immediately obtain the result above. □

**Lemma B.3.** *Let $\lambda \geq 0$ be the regularization parameter, $n$ be the training sample size and $\boldsymbol{\theta}_{\mathrm{ridge}}(N; \lambda)$ be the output of ridge regression. Then*

$$\Delta(\boldsymbol{\theta}_{\mathrm{ridge}}(N; \lambda)) \leq \mathrm{RidgeBias} + \mathrm{RidgeVariance},$$

*where,*

$$\mathrm{RidgeBias} \lesssim \left( \frac{\lambda + \sum_{i > k^*_{\mathrm{ridge}}} \mu_i}{N} \right)^2 \cdot \|\boldsymbol{\theta}^*\|^2_{\mathbf{M}^{-1}_{0:k^*_{\mathrm{ridge}}}}$$
$$+ \|\boldsymbol{\theta}^*\|^2_{\mathbf{M}_{k^*_{\mathrm{ridge}}:\infty}},$$

$$\mathrm{RidgeVariance} \lesssim \sigma^2 \cdot \left\{ \frac{k^*_{\mathrm{ridge}}}{N} + \frac{N \sum_{i > k^*_{\mathrm{ridge}}} \mu_i^2}{\left( \lambda + \sum_{i > k^*_{\mathrm{ridge}}} \mu_i \right)^2} \right\},$$

*where $k^*_{\mathrm{ridge}} := \min \left\{ k : b\mu_{k+1} \leq (\lambda + \sum_{i>k} \mu_i)/n \right\}$.*

*Proof.* The argument is similar to the lower bound above. Since the choice of $k^*_{\mathrm{ridge}}$ is to ensure the sharpness of the upper bound, we can relax this condition to an arbitrary $k$ in $[d]$ and still maintain a valid upper bound. $\qquad\square$

## C. Proof of the Effect of Graph Structure

In this section, we present a proof for the Theorem 4.3. We decompose the proof for Theorem 4.3 into two parts: 1) for the power-law graph and 2) for the regular graph. We start with restating the results of Theorem 4.3 into two corresponding lemmas and proof.

**Lemma C.1.** *Consider a power-law graph $\mathcal{G}_{\mathrm{p}}$ with graph matrix $\mathbf{G}_{\mathrm{p}}$ whose eigen-spectrum is characterized by Eq. 4.1 with a large enough $\beta$. Then for every $\lambda$ for ridge regression, there exists a choice for $\gamma^*$ for SGD such that for sufficiently large $N$, we have*

$$\Delta(\boldsymbol{\theta}_{\mathrm{sgd}}(N, \mathbf{G}_{\mathrm{p}}; \gamma^*)) \lesssim \Delta(\boldsymbol{\theta}_{\mathrm{ridge}}(N, \mathbf{G}_{\mathrm{p}}; \lambda)).$$

*Proof.* By Theorem 4.1, we know the excessive risk of SGD under our setting is given by the following,

$$\mathrm{SGDRisk} \lesssim \underbrace{\frac{1}{\eta^2 N^2} \cdot \left\| \exp(-N\eta\mathbf{M})\boldsymbol{\theta}^* \right\|^2_{\mathbf{M}^{-1}_{0:k_1}} + \|\boldsymbol{\theta}^*\|^2_{\mathbf{M}_{k_1:\infty}}}_{\mathrm{SGDBias}}$$
$$+ \underbrace{(\sigma^2 + \|\boldsymbol{\theta}^*\|_{\widetilde{\mathbf{H}}}) \cdot \left( \frac{k_2}{N} + N\eta^2 \sum_{i > k_2} \mu_i^2 \right)}_{\mathrm{SGDVariance}}. \tag{C.1}$$

where the parameter $k_1, k_2 \in [d]$ can be arbitrarily chosen.

Then recall the lower of the risk achieved by ridge regression with parameter $\lambda$:

$$\mathrm{RidgeRisk} \gtrsim \underbrace{\frac{\widehat{\lambda}^2}{N^2} \cdot \|\boldsymbol{\theta}^*\|^2_{\mathbf{M}^{-1}_{0:k^*}} + \|\boldsymbol{\theta}^*\|^2_{\mathbf{M}_{k^*:\infty}}}_{\mathrm{RidgeBias}} +$$
$$\underbrace{\sigma^2 \cdot \left( \frac{k^*}{N} + \frac{N}{\widehat{\lambda}^2} \sum_{i > k^*} \mu_i^2 \right)}_{\mathrm{RidgeVariance}}, \tag{C.2}$$

where $\widehat{\lambda} = \lambda + \sum_{i > k^*} \mu_i$ and $k^* = \min\{k : N\mu_k \leq \widehat{\lambda}\}$.

For the following analysis, we set $k_1, k_2 = k^*$ for the excessive risk of SGD and divide the analysis into bias and variance.

**Bias.** By (C.1) The bias of SGD is given as follows,

$$\text{SGDBias} \lesssim \frac{1}{\eta^2 N^2} \cdot \left\| \exp(-N\eta \mathbf{M})\boldsymbol{\theta}^* \right\|_{\mathbf{M}_{0:k^*}^{-1}}^2 + \left\| \boldsymbol{\theta}^* \right\|_{\mathbf{M}_{k^*:\infty}}^2$$

From the equation above, we can observe that the bias of SGD can be decomposed into two intervals: 1) $i \leq k^*$ and 2) $i > k^*$.

We start with the second interval. For $i > k^*$, we have that,

$$\text{SGDBias}[k^* : \infty] = \left\| \boldsymbol{\theta}^* \right\|_{\mathbf{M}_{k^*:\infty}}^2$$
$$= \text{RidgeBias}[k^* : \infty].$$

For $i \leq k^*$, the order of the eigenvalue and eigenvectors are preserved and we can decompose each term of bias as follows,

$$\text{SGDBias}[i] = (\boldsymbol{\theta}^*[i])^2 \frac{1}{N^2 \eta^2 \mu_i} \exp\left( - 2\eta N \mu_i \right)$$

Similarly, we can decompose each term of the bias as of Ridge as,

$$\text{RidgeBias}[i] = \frac{\widehat{\lambda}^2}{N} \frac{1}{\mu_i} (\boldsymbol{\theta}[i]^*)^2$$

Now, we can divide the analysis into two cases: **Case I:** $\widehat{\lambda} \geq \text{tr}(\mathbf{M})$ and **Case II:** $\widehat{\lambda} \leq \text{tr}(\mathbf{M})$.

For **Case I:**, we can pick $\eta = 1/\widehat{\lambda}$ and obtain that that,

$$\text{SGDBias}[i] = (\boldsymbol{\theta}^*[i])^2 \frac{\widehat{\lambda}^2}{N^2 \mu_i} \exp\left( - 2\widehat{\lambda} N \mu_i \right)$$
$$= \text{RidgeBias}[i] \exp\left( - 2\widehat{\lambda} N \mu_i \right)$$

because $2\widehat{\lambda} N \mu_i > 0$, we have that,

$$\leq \text{RidgeBias}[i]$$

For **Case II:**, we can pick $\eta = 1/\text{tr}(\mathbf{M})$ and obtain that that,

$$\text{SGDBias}[i] = (\boldsymbol{\theta}^*[i])^2 \frac{\text{tr}(\mathbf{M})^2}{N^2 \mu_i} \exp\left( - 2N\mu_i / \text{tr}(\mathbf{M}) \right)$$
$$= (\boldsymbol{\theta}^*[i])^2 \frac{\widehat{\lambda}^2 \text{tr}(\mathbf{M})^2}{\widehat{\lambda}^2 N^2 \mu_i} \exp\left( - 2N\mu_i / \text{tr}(\mathbf{M}) \right)$$
$$= \text{RidgeBias}[i] \frac{\text{tr}(\mathbf{M})^2}{\widehat{\lambda}^2} \exp\left( - 2N\mu_i / \text{tr}(\mathbf{M}) \right)$$

By definition of $\widehat{\lambda}$, we have that $\widehat{\lambda} \lesssim N\mu_{k^*}$ and can obtain that

$$\lesssim \text{RidgeBias}[i] \frac{\text{tr}(\mathbf{M})^2}{(N\mu_{k^*})^2} \exp\left( - 2N\mu_i / \text{tr}(\mathbf{M}) \right)$$

By the choice of model $\frac{1}{i^\beta}$, as $\beta$ increase, we have that $\text{tr}(\mathbf{M})/\mu_{k^*}$ decrease. Therefore, we can set $\beta$ large enough so that

$$\frac{\text{tr}(\mathbf{M})^2}{(N\mu_{k^*})^2} \leq \exp\left( 2N\mu_i / \text{tr}(\mathbf{M}) \right).$$

Then, we can arrive that,

$$\text{SGDBias[i]} \lesssim \text{RidgeBias}[i] \frac{\text{tr}(\mathbf{M})^2}{(N\mu_{k^*})^2} \exp\left( - 2N\mu_i/\text{tr}(\mathbf{M}) \right),$$
$$\lesssim \text{RidgeBias}[i].$$

Therefore, combining the results above, we have that

$$\text{SGDBias} \lesssim \text{RidgeBiasBoud}.$$

Next, let's consider variance. Again, by the excessive risk upper bound of SGD, we have that

$$\text{SGDVariance} = \left(1 + \frac{\|\boldsymbol{\theta}\|_{\mathbf{M}}^2}{\sigma^2}\right) \cdot \sigma^2 \left(\frac{k^*}{N} + N\eta^2 \sum_{i>k^*} \mu_i^2\right)$$

Similar to the bias analysis, we can divide the analysis into two cases: **Case I:** $\widehat{\lambda} \geq \text{tr}(\mathbf{M})$ and **Case II:** $\widehat{\lambda} \leq \text{tr}(\mathbf{M})$.

For **Case I:** $\widehat{\lambda} \geq \text{tr}(\mathbf{M})$, we pick $\eta = 1/\widehat{\lambda}$ as for the bias:,

$$\text{SGDVariance} = \left(1 + \frac{\|\boldsymbol{\theta}\|_{\mathbf{M}}^2}{\sigma^2}\right) \cdot \sigma^2 \left(\frac{k^*}{N} + \frac{N}{\widehat{\lambda}^2} \sum_{i>k^*} \mu_i^2\right)$$

substitute the premise that $\frac{\|\boldsymbol{\theta}\|_{\mathbf{M}}^2}{\sigma^2} = \Theta(1)$:

$$\lesssim \Theta(1) \cdot \sigma^2 \left(\frac{k^*}{N} + \frac{N}{\widehat{\lambda}^2} \sum_{i>k^*} \mu_i^2\right)$$
$$= \Theta(1) \cdot \text{RidgeVariance}$$
$$\lesssim \text{RidgeVariance}$$

For **Case II:** $\widehat{\lambda} \leq \text{tr}(\mathbf{M})$:, we can pick $\eta = 1/\text{tr}(\mathbf{M})$ as for the bias and obtain that

$$\text{SGDVariance} = (1 + \frac{\|\boldsymbol{\theta}\|_{\mathbf{M}}^2}{\sigma^2}) \cdot \sigma^2 \left(\frac{k^*}{N} + \frac{N}{\text{tr}(\mathbf{M})^2} \sum_{i>k^*} \mu_i^2\right)$$
$$\leq (1 + \frac{\|\boldsymbol{\theta}\|_{\mathbf{M}}^2}{\sigma^2}) \cdot \sigma^2 \left(\frac{k^*}{N} + \frac{N}{\widehat{\lambda}^2} \sum_{i>k^*} \mu_i^2\right)$$

Similarly, by the premise :

$$\lesssim \Theta(1) \cdot \sigma^2 \left(\frac{k^*}{N} + \frac{N}{\widehat{\lambda}^2} \sum_{i>k^*} \mu_i^2\right)$$
$$= \Theta(1) \cdot \text{RidgeVariance}$$
$$\lesssim \text{RidgeVariance}$$

Therefore, we have that

$$\text{SGDVariance} \lesssim \text{RidgeVariance}$$

Combining all the result above, we have that so long as $\beta$ is large enough, there always exists an $\eta$ such that

$$\text{SGDRisk} \lesssim \text{RidgeRisk}.$$

This completes the proof.

$\square$

**Lemma C.2.** *consider a regular graph $\mathcal{G}_r$ with graph matrix $\mathbf{G}_r$ whose eigen-spectrum is characterized by Eq. 4.1 with a small $\beta$. Then for every choice of $\gamma$ for SGD, there exists a $\lambda^*$ such that,*

$$\Delta(\boldsymbol{\theta}_{\mathrm{ridge}}(N, \mathbf{G}_r; \lambda^*)) \lesssim \Delta(\boldsymbol{\theta}_{\mathrm{sgd}}(N, \mathbf{G}_r; \gamma)).$$

*Proof.* Recall that to show that ridge is comparable with SGD in regular graph is enough to show that there exist a $\beta$ small enough, so that for every $\eta$ for SGD we can always find a $\lambda$ for Ridge to achieve

$$\mathrm{RidgeRisk} \lesssim \mathrm{SGDRisk}.$$

Similarly, by Theorem 4.1, we have the excess risk lower bound of SGD given by,

$$\mathrm{SGDRisk} \gtrsim \underbrace{\frac{1}{\eta^2 N^2} \cdot \left\| \exp\left(-N\eta\mathbf{M}\right)\boldsymbol{\theta}^* \right\|^2_{\mathbf{M}_{0:k^*}^{-1}} + \left\|\boldsymbol{\theta}^*\right\|^2_{\mathbf{M}_{k^*:\infty}}}_{\mathrm{SGDBias}}$$

$$+ \underbrace{\frac{\sigma^2}{N} \cdot \left(k^* + N^2\eta^2 \sum_{i>k^*} \mu_i^2\right) + \|\boldsymbol{\theta}^*\|^2_{\mathbf{M}} \frac{\eta}{\mu_1} \sum_{i>k^\dagger} \mu_i^2}_{\mathrm{SGDVariance}},$$

where $\mu_1, \ldots \mu_d$ are sorted eigenvalues for $\mathbf{M}$, $k^* = \max\{k : \widetilde{\lambda}_k \geq 1/(N\eta)\}$, and $k^\dagger = \max\{k : \widetilde{\lambda}_k \geq 2/(3N\eta)\}$.

Similarly, by Theorem 4.2, we have the excess risk upper bound of ridge given by,

$$\mathrm{RidgeRisk} \lesssim \underbrace{\left(\frac{\widehat{\lambda}}{N}\right)^2 \cdot \|\boldsymbol{\theta}^*\|^2_{\mathbf{M}_{0:k^*_{\mathrm{ridge}}}^{-1}} + \|\boldsymbol{\theta}^*\|^2_{\mathbf{M}_{k^*_{\mathrm{ridge}}:\infty}}}_{\mathrm{RidgeBias}}$$

$$+ \underbrace{\sigma^2 \cdot \left(\frac{k^*_{\mathrm{ridge}}}{N} + \frac{N \sum_{i>k^*_{\mathrm{ridge}}} \widehat{\lambda}_i^2}{\widehat{\lambda}^2}\right)}_{\mathrm{RidgeVariance}},$$

where $\mu_1, \ldots, \mu_d$ are the sorted eigenvalues for $\mathbf{M}$ in descending order, and

$$k^*_{\mathrm{ridge}} := \min\left\{k : b\lambda_{k+1} \leq (\lambda + \sum_{i>k} \lambda_i)/n\right\}.$$

Similar to the previous proof, we can decompose the risk profile into bias and variance and then compare them separately.

We start with aligning the risk profiles of SGD and ridge by picking the ridge regression regularization

$$\lambda^* = \frac{b}{\eta} - \sum_{i>k^*} \mu_i.$$

Doing so leads to $k^*_{\mathrm{ridge}} = k^*$.

**Variance.** Then we start by comparing the variance. We denote,

$$\widehat{\lambda}^* = \lambda^* + \sum_{i>k^*} \mu_i.$$

By the choice of $\lambda^*$, we have that,

$$\widehat{\lambda}^* \approx \frac{1}{\eta}.$$

Then, we have the variance for Ridge as,

$$\text{RidgeVariance} \lesssim \sigma^2 \cdot \left( \frac{k^*}{N} + \frac{N}{(\widehat{\lambda}^*)^2} \sum_{i > k^*} \mu_i^2 \right)$$

$$\lesssim \frac{\sigma^2}{N} \cdot \left( k^* + N^2 \eta^2 \sum_{i > k^*} \mu_i^2 \right)$$

$$\leq \frac{\sigma^2}{N} \cdot \left( k^* + N^2 \eta^2 \sum_{i > k^*} \mu_i^2 \right)$$

$$+ \|\boldsymbol{\theta}^*\|_{\mathbf{M}}^2 \frac{\eta}{\mu_1} \sum_{i > k^\dagger} \mu_i^2,$$

$$= \text{SGDVariance}$$

Therefore, we have that

$$\text{RidgeVariance} \lesssim \text{SGDVariance}$$

Next, we focus on the bias. Under same set up as the variance, the Ridge bias is given by,

$$\text{RidgeBias} \lesssim \left( \frac{\widehat{\lambda}^*}{N} \right)^2 \cdot \|\boldsymbol{\theta}^*\|_{\mathbf{M}_{0:k^*}^{-1}}^2 + \|\boldsymbol{\theta}^*\|_{\mathbf{M}_{k^*:\infty}}^2$$

**Bias.** Similar to the previous analysis, we can decompose the bias of Ridge into two intervals: 1) $i \leq k^*$ and 2) $i > k^*$. We start with the second interval. For $i > k^*$,, we immediately obtain,

$$\text{RidgeBias}[k^* : \infty] = \|\boldsymbol{\theta}^*\|_{\mathbf{M}_{k^*:\infty}}^2$$

$$= \text{SGDBias}[k^* : \infty].$$

For $i \leq k^*$, similar to the previous analysis, we decompose each term of bias bound as follows,

$$\text{RidgeBias}[i] = (\boldsymbol{\theta}^*[i])^2 \frac{\widehat{\lambda}^2}{N^2 \mu_i},$$

$$= (\boldsymbol{\theta}^*[i])^2 \frac{\widehat{\lambda}^2}{N^2 \mu_i} \frac{\eta^2}{\eta^2}$$

$$\cdot \exp\left( -2\eta N \mu_i \right) \exp\left( 2\eta N \mu_i \right),$$

$$= (\boldsymbol{\theta}^*[i])^2 \frac{1}{N^2 \mu_i \eta^2}$$

$$\cdot \exp\left( -2\eta N \mu_i \right) \widehat{\lambda}^2 \eta^2 \exp\left( 2\eta N \mu_i \right),$$

$$= \text{SGDBias[i]} \widehat{\lambda}^2 \eta^2 \exp\left( 2\eta N \mu_i \right)$$

Similar to the analysis of variance, we have that,

$$\widehat{\lambda}^* \approx \frac{1}{\eta},$$

and consequently, we have,

$$\widehat{\lambda}^* \eta \approx \Theta(1).$$

Then, we have that,

$$\text{RidgeBias}[i] = \text{SGDBias[i]} \widehat{\lambda}^2 \eta^2 \exp\left( 2\eta N \mu_i \right)$$

$$\lesssim \text{SGDBias[i]} \exp\left( 2\eta N \mu_i \right)$$

By definition of $\widehat{\lambda}^*$, we have that,

$$\eta N \mu_i \leq \eta \widehat{\lambda}^*$$
$$\lesssim \Theta(1).$$

Then, we arrive

$$\mathrm{RidgeBias}[i] \lesssim \mathrm{SGDBias[i]}$$

$$\mathrm{RidgeBias} \lesssim \mathrm{SGDBias}$$

Combining all the results above, we have that there exists an $\lambda$ such that

$$\mathrm{RidgeRisk} \lesssim \mathrm{SGDRisk}.$$

$\square$

Then, combining the result from the two lemmas above, we immediate obtain a proof for Theorem 4.3.

## D. Stacking GNN Layers

In this appendix, we present a proof for Proposition 4.5 and a further discussion on the over-smoothing of GNNs.

*Proof of Proposition 4.5.* First recall that the definition of $\widehat{\mathbf{G}}(l)$ is given by,

$$\widehat{\mathbf{G}}(l) = \prod_{i=1}^{l} \mathbf{G}.$$

Without loss of generality, we may assume that

$$\mu_i(\mathbf{G}) \geq \mu_j(\mathbf{G}).$$

Then, by the definition above, we have that,

$$\frac{\mu_i(\widehat{\mathbf{G}}(l+1))}{\mu_j(\widehat{\mathbf{G}}(l+1))} = \frac{\mu_i(\prod_{i=1}^{l+1} \mathbf{G})}{\mu_j(\prod_{i=1}^{l+1} \mathbf{G})}$$
$$= \frac{\mu_i(\prod_{i=1}^{l} \mathbf{G})}{\mu_j(\prod_{i=1}^{l} \mathbf{G})} \cdot \frac{\mu_i(\mathbf{G})}{\mu_j(\mathbf{G})}$$
$$= \frac{\mu_i(\widehat{\mathbf{G}}(l))}{\mu_j(\widehat{\mathbf{G}}(l))} \cdot \frac{\mu_i(\mathbf{G})}{\mu_j(\mathbf{G})}$$

Then, by Assumption 3.4 and the premise that

$$\mu_i(\mathbf{G}) \geq \mu_j(\mathbf{G}),$$

then we have that,

$$\frac{\mu_i(\widehat{\mathbf{G}}(l+1))}{\mu_j(\widehat{\mathbf{G}}(l+1))} = \frac{\mu_i(\widehat{\mathbf{G}}(l))}{\mu_j(\widehat{\mathbf{G}}(l))} \cdot \frac{\mu_i(\mathbf{G})}{\mu_j(\mathbf{G})}$$
$$\geq \frac{\mu_i(\widehat{\mathbf{G}}(l))}{\mu_j(\widehat{\mathbf{G}}(l))} \cdot 1$$
$$= \frac{\mu_i(\widehat{\mathbf{G}}(l))}{\mu_j(\widehat{\mathbf{G}}(l))}.$$

$\square$

# E. Extended Related Work

In this section, we provide a comprehensive overview of key literature closely related to the theoretical analysis presented in this paper. Specifically, we discuss three primary areas: (1) theoretical understanding of Graph Neural Networks (GNNs), (2) excessive risk analysis of learning algorithms, and (3) spectral graph theory.

**Theoretical Understanding of GNNs.** Due to the empirical success of GNNs, there is a growing body of theoretical work that investigates their expressive power and generalization capability. The expressive power of GNNs, referring to their capacity to distinguish different graph structures, is often assessed by comparing GNN models to classical graph isomorphism tests, such as the Weisfeiler-Lehman (WL) test (Leman & Weisfeiler, 1968; Jegelka, 2022; Sato, 2020; Zhang et al., 2023a;b; Xu et al., 2018). Generalization analyses explore how GNNs perform on unseen data, under different architecture (Tang & Liu, 2023) or using complexity measures (e.g., VC-dimension, Rademacher complexity)(Lv, 2021; Ma et al., 2021; Liao et al., 2021), Neural Tangent Kernel (NTK)(Du et al., 2019), and information-theoretic tools (mutual information, entropy) (Verma & Zhang, 2019; Zhu et al., 2021). These studies primarily focus on factors like architecture and input graph characteristics, and remain orthogonal to our analysis. In addition, there is a recent works that tries to establish a theoretical connection between the topology-awareness (how well a graph structure is captured) of GNN and its generalization performance (Su & Wu, 2024).

A related line of research examines the convergence behavior of GNN learning algorithms (Chen et al., 2017; Huang et al., 2018; Chen et al., 2018; Awasthi et al., 2021; Li et al., 2018; Oono & Suzuki, 2020). Such studies typically provide convergence rates but remain restricted to the interpolation (noise-free) setting and offer limited insights into how graph structures influence algorithmic performance. It remains largely unexplored how different graph characteristics (e.g., power-law vs. regular) systematically affect the generalization performance of algorithms when noise is present. Our work addresses precisely this gap, providing a novel framework linking graph structure explicitly to the generalization performance of learning algorithms such as SGD and Ridge regression.

Furthermore, a significant practical challenge in GNN design includes issues of oversmoothing (Rusch et al., 2023) and oversquashing (Topping et al., 2021). Oversmoothing occurs when deep message-passing leads to indistinguishable node representations, reducing model performance in tasks that require distinct embeddings. Oversquashing refers to the over-compression of information from distant nodes, impeding long-range dependency capture in deep networks. Our theoretical analysis of learning algorithm behavior with respect to graph structure can also provide new perspectives and analytical tools for investigating such phenomena, particularly oversmoothing.

**Excessive Risk of Learning Algorithms.** The analysis of excessive risk is central to learning theory literature (Dhillon et al., 2013; Lakshminarayanan & Szepesvari, 2018; Jain et al., 2017; Zou et al., 2021b; 2023; Tsigler & Bartlett, 2023). Extensive research investigates the generalization properties of classic learning algorithms, notably Stochastic Gradient Descent (SGD) and Ridge regression. Non-asymptotic risk bounds have been thoroughly characterized for SGD and Ridge in both under- and over-parameterized regimes (Hsu et al., 2012; Kobak et al., 2020; Dieuleveut et al., 2017; Bach & Moulines, 2013; Jain et al., 2018; Défossez & Bach, 2015; Paquette et al., 2022; Tsigler & Bartlett, 2023). For instance, constant-stepsize SGD with tail-averaging achieves minimax-optimal rates for least-squares tasks (Jain et al., 2017). Recent works also provided detailed excess-risk characterizations for Ridge regression, depending critically on spectral properties of data covariance matrices (Dobriban & Wager, 2018; Hastie et al., 2022; Wu & Xu, 2020; Xu & Hsu, 2019). Nevertheless, how these theoretical insights extend to graph learning scenarios, particularly within GNN frameworks, remains largely unexplored. We expand this knowledge by explicitly analyzing and comparing the excessive risk of SGD and Ridge regression in graph-structured settings, thereby shedding new light on the impact of structural characteristics on learning algorithm performance.

**Spectral Graph Theory.** Spectral graph theory studies graph properties via eigenvalues and eigenvectors of associated matrices (e.g., adjacency, Laplacian matrices)(Chung, 1997; Spielman, 2012; Pósfai & Barabási, 2016; Van Mieghem, 2023; Gera et al., 2018). Foundational results, including the Perron-Frobenius theorem and Cheeger's inequality, provide insights into the connectivity, robustness, and spectral characteristics of graphs. For example, power-law graphs (characterized by heterogeneous degree distributions and heavy-tailed eigenspectra) differ markedly from regular graphs (uniform degree distributions and evenly spaced eigenspectra)(Faloutsos et al., 1999; Chung et al., 2003; Farkas et al., 2001; Goh et al., 2001; Easley et al., 2010). These contrasting spectral profiles are essential for interpreting network structure, influencing information propagation dynamics, robustness, and learning behavior. Our theoretical framework leverages spectral graph

theory to explicitly relate eigenvalue decay patterns (e.g., power-law versus uniform) to the performance characteristics of learning algorithms. Thus, spectral graph theory forms a foundational tool in our analysis, directly linking structural properties of graphs to algorithmic performance differences.

Overall, this paper synthesizes these theoretical domains—GNN expressivity and generalization, excessive risk of algorithms, and spectral graph theory—to provide new insights into the interplay between graph structures and learning algorithms. Our work represents a step towards a deeper theoretical understanding of graph-based learning, offering tools and analyses applicable broadly across various graph learning contexts.

# F. Experiment Details

In this appendix, we present additional details on our experimental study. This appendix focuses on additional details on the testbed, graph generation model and hyper-parameter search.

## F.1. Testbed

Our experiments were conducted on a Dell PowerEdge C4140, The key specifications of this server, pertinent to our research, include:
**CPU:** Dual Intel Xeon Gold 6230 processors, each offering 20 cores and 40 threads.
**GPU:** Four NVIDIA Tesla V100 SXM2 units, each equipped with 32GB of memory, tailored for NV Link.
**Memory:** An aggregate of 256GB RAM, distributed across eight 32GB RDIMM modules.
**Storage:** Dual 1.92TB SSDs with a 6Gbps SATA interface.
**Networking:** Features dual 1Gbps NICs and a Mellanox ConnectX-5 EX Dual Port 40/100GbE QSFP28 Adapter with GPUDirect support.
**Operating System:** Ubuntu 18.04LTS.

## F.2. Graph Generation Model

For our experiment, we rely on the NetworkX python (Hagberg et al., 2008) library. In particular, we use the default implementation of regular graph generation and Barabasi-Albert model (Pósfai & Barabási, 2016) from the Networkx library. The Barabási-Albert (BA) model is a widely used generative model for creating scale-free networks, which are networks characterized by a power-law degree distribution. The core idea behind the BA model is to capture the "preferential attachment" mechanism, where new nodes are more likely to connect to existing nodes that already have a high degree of connections. This reflects many real-world networks, such as social networks, where popular individuals (nodes) tend to attract more connections.

**Rough Procedure of Barabási-Albert model.** The Barabási-Albert model generates a network through the following steps:

1. **Initialization:** Start with a small connected network of $m_0$ nodes.

2. **Growth:** Add one new node at a time. Each new node forms $m$ edges that link it to $m$ existing nodes.

3. **Preferential Attachment:** The probability that a new node will connect to an existing node $i$ is proportional to the degree of node $i$. Formally, the probability $P(i)$ that the new node connects to node $i$ is given by:

$$P(i) = \frac{k_i}{\sum_j k_j}$$

where $k_i$ is the degree of node $i$ and the sum is over all existing nodes.

**Hyperparameters** The BA model has two key hyperparameters:

- **$m_0$** : The initial number of nodes in the network.

- **m** : The number of edges that each new node will add when it is introduced to the network. This parameter influences the density and the structure of the resulting network.

These hyperparameters directly affect the network's topology and are crucial in determining the characteristics of the scale-free network generated by the BA model. We adopt the default $\mathbf{m_0}$ from the implementation of NetworkX library, and set $\mathbf{m}$ to be 3, which is a commonly used value to model real-life network.

### F.3. Hyper Parameter Search for Learning Algorithm

The main hyper-parameter for SGD is the learning rate $\eta$. In the statement of the results for the excessive risk of SGD, there is a requirement for the learning that relates to the eigenspectrum of the data covariance. For our experiment, we have access to the full information of the data covariance. As such, we simply infer a possible range for the learning rate and do a grid research for the learning rate in the range. The search process is done through an independent (validation) dataset. Therefore, there is no interference to the testing process.

Similarly, the main hyper-parameter for Ridge is the regularization parameter $\lambda$. For $\lambda$, there is no theoretical guidance from the analysis on its possible values. Therefore, we do a grid research on a large array of $\lambda$ values ranging from $[0.1, 1000]$. The research process is similar to the one for the learning rate.

## G. Additional Discussion

In this section, we present some additional discussion on the formulation of the problem.

### G.1. Problem Formulation and GNNs

**Message-Passing Graph Neural Networks.** The computation in GNNs can be viewed as message-passing along graph structure (Jegelka, 2022). At each round $l$, the new embedding $\mathbf{x}_i^{(l)}$ for vertex $i$ is updated through a series of aggregate and combine steps as outlined below:

$$\mathbf{m}_i^{(l)} = \text{AGGREGATE}(\{\mathbf{x}_j^{(l-1)} \in \mathcal{N}(i)\}),$$
$$\mathbf{x}_i^{(l)} = \text{COMBINE}(\mathbf{x}_i^{(l-1)}, \mathbf{m}_i^{(l)}),$$

where $x_i^{(0)}$ is initialized as the feature vector $\mathbf{x}_i$, $\mathcal{N}(i)$ represents the neighbors of vertex $i$, and $\mathbf{m}_i$ denotes the aggregated representation at round $l$. Different GNNs differ in specific implementation of the AGGREGATE and COMBINE functions. Essentially, a GNN serves as an embedding function that integrates the graph structure and node features to produce an aggregated representation vector of node $v$. This representation is subsequently processed by a read-out function (e.g., a ReLU layer) to generate predictions.

**One-round Graph Neural Network.** In this study, we follow a similar setting to (Awasthi et al., 2021) and focus our main discussions on a one-round GNN that consists of an aggregation operation followed by a readout operation. We interpret the aggregation step as a graph matrix $\mathbf{G}$ acting upon the feature matrix $\mathbf{X}$. Formally, we decompose the one-round GNN into two distinct components. The first component aggregates node features via a specified aggregation operator and the chosen graph matrix $\mathbf{G}$, producing an intermediate representation space denoted by $\mathcal{M}$:

$$\mathcal{M} = \mathbf{G} \circ \mathbf{X}. \tag{G.1}$$

Here, $\mathcal{M}$ represents the intermediate aggregated feature space, and we assume without loss of generality that it is a subspace of a Hilbert space $\mathcal{H}$. Different choices of the aggregation matrix $\mathbf{G}$ and the aggregation operation $\circ$ allow us to recover various canonical graph neural network architectures. Below, we provide a comprehensive discussion of several prominent examples:

- **Graph Convolutional Networks (GCN).** When the graph matrix $\mathbf{G}$ is chosen as the symmetrically normalized adjacency matrix $\widehat{\mathbf{A}} = \widetilde{\mathbf{D}}^{-1/2}\widetilde{\mathbf{A}}\widetilde{\mathbf{D}}^{-1/2}$, where $\widetilde{\mathbf{A}} = \mathbf{A} + \mathbf{I}$ and $\widetilde{\mathbf{D}}$ is the degree matrix of $\widetilde{\mathbf{A}}$, and the aggregation operator $\circ$ is simple matrix multiplication, we recover the standard GCN formulation proposed by (Kipf & Welling, 2017):

$$\mathcal{M}_{\text{GCN}} = \widehat{\mathbf{A}}\mathbf{X}. \tag{G.2}$$

- **GraphSAGE (Mean Aggregation).** When the graph matrix $\mathbf{G}$ corresponds to a row-normalized adjacency matrix $\widetilde{\mathbf{A}}\mathrm{rw} = \widetilde{\mathbf{D}}^{-1}\widetilde{\mathbf{A}}$, where $\widetilde{\mathbf{A}}$ again denotes adjacency with self-loops, we recover the mean-aggregation variant of Graph-SAGE (Hamilton et al., 2018). Specifically:

$$\mathcal{M}_{\mathrm{SAGE}} = \widetilde{\mathbf{A}}_{\mathrm{rw}}\mathbf{X}. \tag{G.3}$$

  Here, each node's representation is updated using the average of its neighbors' features, incorporating self-information as well.

- **Graph Attention Networks (GAT).** For Graph Attention Networks (Veličković et al., 2017), the aggregation is achieved by a learned attention mechanism rather than fixed normalization. Here, the aggregation operator $\circ$ represents element-wise multiplication weighted by attention scores, resulting in a data-dependent and adaptive $\mathbf{G}(\mathbf{X})$:

$$\mathcal{M}_{\mathrm{GAT}} = \mathbf{G}_{\mathrm{att}}(\mathbf{X}) \circ \mathbf{X}, \tag{G.4}$$

  where $\mathbf{G}_{\mathrm{att}}(\mathbf{X})$ encodes attention-based weights, dynamically computed through a neural attention mechanism.

- **Simplified Graph Convolution (SGC).** By extending the GCN aggregation step to multiple rounds but merging linear operations into a single step, the simplified graph convolutional network (SGC) (Wu et al., 2019) is recovered. Formally, if $\mathbf{G}$ is the $K$-th power of $\widehat{\mathbf{A}}$, we have:

$$\mathcal{M}_{\mathrm{SGC}} = \widehat{\mathbf{A}}^K \mathbf{X}. \tag{G.5}$$

  This corresponds to applying multiple rounds of smoothing/aggregation in a simplified, efficient manner without intermediate nonlinearities.

This comprehensive discussion demonstrates how the choice of graph matrix $\mathbf{G}$ and aggregation operation $\circ$ yields various canonical GNN variants, each capturing different relational structures and neighborhood aggregation strategies.

