# OpenReview forum: "On the Interplay between Graph Structure and Learning Algorithms in Graph Neural Networks"
_ICML.cc/2025/Conference — ICML 2025 poster_

### Official Review · Reviewer_8EPb · 2025-03-08

**Overall Recommendation:** 4

**Summary:**

This paper explores the relationship between graph structure and learning algorithms in Graph Neural Networks (GNNs), particularly in the context of generalization performance. Unlike prior work that primarily focuses on convergence rates in noise-free settings, this study extends those analysis to scenarios with noise, providing a more comprehensive understanding of learning dynamics in GNNs. To me, the paper's main contributions include:

1. Theoretical Analysis of Excess Risk: The authors derive excess risk profiles for two learning algorithms, i.e., Stochastic Gradient Descent (SGD) and Ridge regression, when applied to GNNs. By leveraging spectral graph theory, they manage to establish a direct connection between excess risk and the underlying graph structure.

2. Impact of Graph Structure: The study systematically compares different types of graph structures, particularly regular graphs and power-law graphs, to determine their effects on learning performance. The analysis reveals that graph topology plays an important role in influencing the behavior of learning algorithms in GNNs.

3. Insights on Multi-Layer GNNs and Over-Smoothing: The authors uncover a growing non-isotropic effect in excess risk, which leads to a new perspective on the well-known over-smoothing issue in deep GNNs.

4. Empirical Validation: Their theoretical findings are further supported by empirical experiments.

Overall, the paper provides a rigorous theoretical framework that deepens our understanding of how learning algorithms interact with graph structures in GNNs. The results can have implications for both theory and practice.

##update after rebuttal

**Claims And Evidence:**

The theoretical contributions of the paper are presented rigorously -- several theorems and propositions outlining the connections between excess risk and graph structure. While the full proofs are given in the appendix, their proof sketches appear correct to me. Moreover, the paper includes empirical results that validate the theoretical predictions. The experiments are designed to test key claims, such as the influence of graph structure on excess risk and the non-isotropic effects in multi-layer GNNs.

**Essential References Not Discussed:**

No. The references seem sufficient to me.

**Experimental Designs Or Analyses:**

For empirical validation, the authors use synthetic graphs with well-controlled structural properties (i.e., regular vs. power-law). While I believe real-world datasets could further support the practical relevance of the findings, the use of synthetic graphs is justified given the study’s theoretical nature.

**Methods And Evaluation Criteria:**

The theoretical analysis of this paper is grounded in spectral graph theory. The choice of excess risk as a key metric is reasonable, as it directly measures the generalization performance of learning algorithms in GNNs.

**Other Comments Or Suggestions:**

N/A

**Other Strengths And Weaknesses:**

N/A

**Questions For Authors:**

I have the following two questions.

1. The main results of the paper rely on the four assumption in Section 3. Please better explain the assumptions and how realistic they are.
2. The paper focuses on regular and power-law graphs. How about random graphs? What results can be expected?

**Relation To Broader Scientific Literature:**

1. This paper bextends several key areas of existing research in graph neural networks, learning theory, and spectral graph theory. Prior work on GNNs has primarily focused on understanding their expressivity, convergence rates, and oversmoothing issues. This paper advances the field by shifting the focus toward generalization performance and excess risk in noisy settings. The authors provide new insights into how graph topology influences learning dynamics in GNNs, complementing prior work that has focused on message passing and feature propagation.

2. In terms of learning theory, previous studies have extensively analyzed the behavior of SGD and Ridge regression in standard machine learning settings. This paper extends these classical results to the GNN setting, providing a novel perspective on how these algorithms interact with graph structures.

3. The analysis of over-smoothing in multi-layer GNNs builds upon existing literature on the depth limitations of GNNs, this paper provides a new theoretical lens on the issue by framing it in terms of non-isotropic excess risk effects.

**Theoretical Claims:**

The proofs seem good me, although I didn't check every step of the proofs in the appendix.

---

> ### Author Rebuttal · Authors · 2025-03-31
>
> Dear Reviewer, thank you for your positive comments and thoughtful questions! These greatly help us in making our paper better, and we appreciate the opportunity to address your questions here.
>
> ## Explanation and Realism of Assumption
>
> Our assumptions are mild and broadly applicable in both theoretical and practical settings.
>
> Assumption 3.1 is a standard and mild assumption in statistical learning theory. Here, $\mathbf{H} = \mathbb{E}[\mathbf{x}\mathbf{x}^\top]$ represents the second moment (or covariance) matrix of the input features. By definition, $\mathbf{H}$ is always positive semidefinite. The condition $\text{tr}(\mathbf{H}) < \infty$ is satisfied so long as the feature distribution has finite energy, which holds for most real-world data, including bounded, Gaussian, and sub-Gaussian/sub-exponential distributions.
>
> Assumption 3.2 is the fourth-moment control of feature vectors. It ensures that the covariance of quadratic forms of $\mathbf{x}$ is not too "wild". This is a standard assumption used in the analysis of generalization, stability, and convergence in learning theory[2,3], and is equivalent to the commonly used bounded second moment assumption of the stochastic gradient in SGD analysis [1]. In particular, this assumption holds for Gaussian, sub-Gaussian and sub-exponential settings common in GNN applications.
>
> Assumption 3.3 states that the noise-weighted covariance $\mathbb{E}[\epsilon^2 \mathbf{x}\mathbf{x}^\top]$ is bounded by $\sigma^2 \mathbf{H}$. This condition ensures that the noise remains controlled relative to the signal, which is a standard assumption in the analysis of both SGD and Ridge regression [2, 3]. It reflects the fundamental learnability of the problem under noisy supervision and guarantees that the problem is well-posed.
>
> Assumption 3.4 requires the graph matrix $\mathbf{G}$ to be symmetric positive definite with bounded norm. This is a practically realistic and widely satisfied condition, as most commonly used graph matrices in GNNs (e.g., adjacency matrix, Laplacian, or their normalized variants) are PSD and have bounded norms.
>
> Together, these assumptions are well-supported by prior work and satisfied by a wide range of data and GNN settings.
>
> ## Random Graphs
>
> Thank you for the interesting question. Our framework and results can indeed be extended to the setting of random graphs.
>
> If we consider individual realizations from a random graph model (e.g., Erdős–Rényi or stochastic block models), our analysis still applies directly, as it is based on the spectrum of the graph matrix, which can be computed for any given instance. In this case, our theoretical results hold conditionally on the observed graph realization.
>
> To generalize our results to the entire random graph model, the spectral properties of the graph matrix would need to be treated in a probabilistic manner. For example, rather than working with deterministic eigenvalues, we would analyze their behavior in expectation or with high probability. In such cases, our bounds and excess risk decomposition could be extended by incorporating concentration inequalities or spectral properties of random matrices.
>
> This is a promising direction for future work, especially since random graph families are expressive and widely used in modeling real-world networks.
>
> Thank you again for your diligent review, and we will incorporate the above discussion in the revision. We hope our response has satisfactorily addressed your questions.
>
> [1] "SGD: General analysis and improved rates." *International conference on machine learning*.
>
> [2] “Benign overfitting in ridge regression.” Journal of  Machine Learning Research.
>
> [3] “Benign overfitting of constant-stepsize sgd for linear regression”. In Conference on Learning Theory

---

### Official Review · Reviewer_ECHh · 2025-03-11

**Overall Recommendation:** 3

**Summary:**

The paper studies how graph structure influences the learning dynamics and generalization performance of GNNs, going beyond traditional noise-free settings to analyze SGD and Ridge regression under moew realistic conditions. Using spectral graph theory, it establishes connections between excess risk and the graph structure, explores the impact of different graph types (regular vs. power-law), and offers guidance for algorithm design and selection.

**Claims And Evidence:**

Evidence needed for the claim "Most real-life graphs, particularly those used for academic benchmarks, are power-law graphs" in line 340-341.

**Essential References Not Discussed:**

The key contribution is to understand the geneneralization of GNNs, but it missed to discuss a paper on the very same topic: Towards Understanding the Generalization of Graph Neural Networks by Tang and Liu in ICML 2023. The paper also discussed how would different GNN architectures affect the generalization bound. I wonder what ditinguishes your result from theirs?

**Experimental Designs Or Analyses:**

Yes. The experimental designs and analyses seem fine to me.

**Methods And Evaluation Criteria:**

Yes

**Other Comments Or Suggestions:**

Besides synthetic data, I wonder if you could verify your result on standard small benchmark datasets, including both homophilic and heterophilic ones?

**Other Strengths And Weaknesses:**

The paper is very well written.

**Questions For Authors:**

1. What is the sepecific G used in Section 4.4? Also I wonder besides graph structure, isn't what architecture one uses also affects the spectrum (say GCN vs GIN).

2. What is the loss function $l$ used in the analysis (line 264-266)? Are you just using the direct difference between the output and the label $y_t$?

**Relation To Broader Scientific Literature:**

Understanding the generalization ability of neural networks is a fundamental problem in learning theory. How would graph structure affects it is a meaningful question to study.

**Theoretical Claims:**

I did not check the proof.

---

> ### Author Rebuttal · Authors · 2025-03-31
>
> Dear Reviewer, thank you for the positive comments and thoughtful questions. These greatly help us in making our paper better, and we appreciate the opportunity to address your concerns and questions here.
>
> ## **Suggested Reference**
>
> Thank you for highlighting this relevant work. While both papers aim to understand the generalization of GNNs, our focus and methodology differ significantly.
>
> Tang and Liu (ICML 2023) study the impact of GNN architectures under the interpolation regime with noiseless labels in a transductive setting. In contrast, our work investigates the interaction between graph structure and learning algorithms (e.g., SGD vs. Ridge) in a noisy, classical generalization setting.
>
> As a result, the two papers adopt different theoretical frameworks—stability and Lipschitz analysis in their case, versus bias-variance decomposition and spectral analysis in ours—and offer complementary insights: theirs on GNN architecture and ours on the interplay between learning algorithm and graph structure.
>
> We will include this reference in the revised paper to enrich the related work section.
>
> ## Specific G used in Section 4.4 and impact of architecture on the spectrum.
>
> Thank you for the great question!
>
> In our empirical study (Section 4.4), we use the normalized adjacency matrix, aligning with the standard GCN setup.
>
> You're absolutely right — the choice of GNN architecture does influence how spectral information is processed. However, our paper focuses on isolating the coupled effect of graph structure and learning algorithms under a fixed architecture. This design choice allows us to clearly attribute differences in generalization behavior to variations in graph structure and learning algorithms, rather than architectural differences. Therefore, the comparison analysis (Theorem 4.3) studies the effect of graph structure with respect to the learning algorithm under an arbitrary fixed architecture.
>
> That said, the theoretical results on excess risk profiles (Theorems 4.1 and 4.2) are general and remain applicable regardless of the specific GNN architecture. Therefore, the core insights of our comparison framework still hold. Extending this framework to jointly analyze how graph structure and architecture together influence the performance of learning algorithm is an exciting direction for future work and can offer a more comprehensive understanding of GNN.
>
> ## Loss function
>
> Thank you for the question. The loss function we use is the standard squared error loss, as specified in lines 170–177. This is a common choice in the learning theory literature [1,2] as well as prior GNN analytical works [3].
>
> ## Additional experiments with benchmark datasets
>
> Thank you for the suggestion. We agree that evaluating on real-world benchmark datasets is valuable. While it is challenging to fully verify all aspects of our theory due to the lack of controllability in real-world data, we conducted additional experiments to assess the robustness of our conclusions on two widely used benchmark datasets with power-law structure: Cora (homophilic) and Chameleon (heterophilic), both available via the DGL library.
>
> For these experiments, we followed the same experimental setup as in the main paper: a regression task was constructed following our analytical framework, and a vanilla GCN was used as the model. According to our theory, so long as the feature matrix satisfies our assumptions, the distinction between homophily and heterophily should not significantly affect the comparative performance of learning algorithms. Moreover, since both datasets exhibit power-law structure, we expect SGD to outperform Ridge.
>
> The results below support this prediction: SGD consistently outperforms Ridge regression on both datasets. This provides further evidence that our theoretical insights hold across a range of graph structures, including both homophilic and heterophilic settings.
>
> Cora
>
> | Sample Size (x10) | 1 | 5 | 10 | 50 | 100 |
> | --- | --- | --- | --- | --- | --- |
> | SGD | 5.68 | 2.67 | 1.32 | 1.08 | 1.03 |
> | Ridge | 9.68 | 3.85 | 1.53 | 1.12 | 1.06 |
>
> Chameleon
>
> | Sample Size (x10) | 1 | 5 | 10 | 50 | 100 |
> | --- | --- | --- | --- | --- | --- |
> | SGD | 8.68 | 3.33 | 1.85 | 1.03 | 0.99 |
> | Ridge | 12.5 | 5.54 | 2.25 | 1.10 | 1.02 |
>
> We will include these results in the revision to further strengthen the empirical support for our theoretical claims.
>
> Thank you again for your diligent review, and we hope our response has satisfactorily addressed your questions and concerns.
>
> [1] “Benign overfitting of constant-stepsize sgd for linear regression”. In Conference on Learning Theory
>
> [2] “Benign overfitting in ridge regression.” Journal of  Machine Learning Research.
>
> [3] "A convergence analysis of gradient descent on graph neural networks." *Advances in Neural Information Processing Systems*

---

### Official Review · Reviewer_ERj6 · 2025-03-13

**Overall Recommendation:** 2

**Summary:**

The authors analyze the excess risk of stochastic gradient descent (SGD) and ridge regression for certain GNNs. Their theoretical analysis, grounded in graph spectral theory and theoretical results from Benign overfitting, demonstrates that SGD can outperform ridge regression on power-law graphs, while ridge regression can be more effective on regular graphs. They validate these findings through experiments on synthetic data.

**Claims And Evidence:**

Some claims/statements are overstated. For example,
- the authors do not analyze GNNs
- The authors do not show clear connections between the graph topology and SGD/Ridge regression, but rather between the spectrum of the second moment of the aggregated features ,which liein  a latent space, and the learning algorithms

**Essential References Not Discussed:**

see below

**Experimental Designs Or Analyses:**

Yes.

**Methods And Evaluation Criteria:**

Yes.

**Other Comments Or Suggestions:**

See above

**Other Strengths And Weaknesses:**

### Strenghts

- The paper addresses an important and interesting question: How does graph topology affect learning? This is a fundamental problem in graph-based learning and has broad implications.
- The results appear convincing and align with experimental observations. In particular, the authors provide clear and practical recommendations on when to use ridge regression versus SGD.
- The paper is well-structured and clearly written, making the theoretical analysis accessible to readers.

### Weaknesses
- The authors claim that their "generating model is very general," but this is misleading. The model does not allow for learnable parameters in graph convolutions, only in the final linear prediction head, meaning no real-world GNNs fall under this framework.
- The assumptions (e.g., 3.1–3.4) are not clearly discussed or justified. It is unclear how restrictive they are or how they impact the conclusions.
- Most of the assumptions and dependencies are formulated in the latent space rather than the graph space. While this simplifies the proofs (by leveraging prior work from non-graph domains), it raises concerns about whether the authors actually answer their original research question regarding graph topology.
- The discussion on oversmoothing seems problematic. Oversmoothing is inherently tied to the spectrum of the graph Laplacian, independent of the learnable algorithm. The authors do not provide sufficient evidence to support their claims in this regard.

### Minor
- The observation that increasing the number of layers can be beneficial is not novel. Prior work has shown that a certain level of smoothing is useful when the ground truth is dominated by low-frequency components (see, e.g., [1]).
- Many works on generalization explicitly depend on graph topological features, such as degree [2] or graph size [3], which the paper does not sufficiently acknowledge.
- The impact statement is missing


[1] Keriven, N. (2022). Not too little, not too much: a theoretical analysis of graph (over) smoothing. Advances in Neural Information Processing Systems, 35, 2268-2281.
[2] Liao, R., Urtasun, R., & Zemel, R. (2020). A pac-bayesian approach to generalization bounds for graph neural networks. arXiv preprint arXiv:2012.07690.
[3] Maskey, S., Kutyniok, G., & Levie, R. (2024). Generalization bounds for message passing networks on mixture of graphons. arXiv preprint arXiv:2404.03473.

**Questions For Authors:**

1. Why do the eigenvalues of $M$ correspond to the graph topology? Could you clarify this connection?
2. How restrictive are assumptions 3.1–3.4? Could you provide concrete examples where they hold in real-world scenarios?
3. In Theorem 4.1, how can $k_1, k_2$​ be chosen arbitrarily? What happens if $k_1$​ is larger than the matrix dimension itself?
4. The proof of the lower bound in Theorem 4.1 is missing. Is $k^∗$ the maximum over all such $k$? Wouldn’t the maximum always be the matrix dimension, or would it be undefined in certain cases?
5. How could your results generalize to actual GNNs used in practice? For example, do you see experimental evidence supporting this claim?

**Relation To Broader Scientific Literature:**

see below

**Theoretical Claims:**

I checked some of the proofs.

---

> ### Author Rebuttal · Authors · 2025-03-31
>
> Dear Reviewer, thank you for thoughtful comments and questions. These greatly help us in making our paper better, and we appreciate the opportunity to address your concerns and questions here. (Please note that the order of our responses may not exactly follow the sequence of your comments.)
>
> ## GNN Formulation and Transferability of Results
>
> We would like to clarify that our formulation is applicable to one-layer GNNs, as also noted by other reviewers. In lines 204–207, we show that it recovers standard GCNs and generalizes the one-layer GNN setting presented in [1]. Furthermore, our empirical studies use standard GCN layers from DGL, and the observed consistency with our theoretical predictions demonstrates that the conclusions are transferable to real-world GNN architectures.
>
> ## Graph Topology and Spectrum of $\mathbf{M}$
>
> Thank you for pointing this out. While $\mathbf{M}$ denotes the covariance of aggregated representations, it is directly shaped by the graph matrix used in the aggregation step. Since the spectrum of graph matrices is known to reflect graph topology, the spectrum of $\mathbf{M}$ inherits key structural properties of the graph. This enables our graph-dependent theoretical results (e.g., Theorem 4.3 and Proposition 4.4). This connection is also empirically illustrated in Figure 2(c,d), where the spectrum of $\mathbf{M}$ reflects structural differences in the graph (e.g., power-law vs. regular).
> ## Suggested Reference
>
> Thank you for pointing out these relevant references. While the cited works share a similar goal of understanding GNN generalization, our paper explores a distinct direction—specifically, the interaction between graph structure and learning algorithm. As a result, our theoretical tools, assumptions, and insights differ significantly. We will include these references in the revised manuscript to enrich the related work section.
>
>
> ## Discussion on Over-smoothing
>
> Thank you for the thoughtful comment. We believe there may be a slight misunderstanding. Our theory does not contradict the established connection between oversmoothing and the spectrum of the graph Laplacian. Rather, our work offers a complementary perspective, focusing on how oversmoothing manifests in model performance, particularly excess risk, as GNN depth increases.
>
> A key insight of our analysis is that the effect of oversmoothing on generalization can also depend on the learning algorithm and the alignment of the ground truth with the spectral components. Specifically, our analysis shows that when the ground truth is concentrated in the head of the eigenspectrum, increasing GNN depth can actually benefit learning algorithms such as SGD, rather than harm performance. This is supported by our empirical results in Figure 3(c,d), where alignment significantly alters the impact of increasing GNN layers under SGD.
>
> We appreciate the opportunity to clarify this and will revise the manuscript to highlight this distinction more clearly.
>
> ## Missing Impact Statement
>
> Thank you for noting this. We will make sure to include the impact statement in the revision. As this is a theoretical study, we do not see any potential negative societal impacts that need to be highlighted.
>
> ## Assumptions 3.1–3.4
>
> Due to space constraints, please refer to our response to Reviewer 8EPb for a detailed discussion of these assumptions.
>
> ## Clarification on Theorem 4.1
>
> Thank you for the questions.
>
> - In line 243, we state that “$k_1,k_2 \in [d]$ are arbitrary,” which means that $k_1$ and $k_2$ can be freely chosen within the valid index range $[1, d]$.
> - The definition of $k^*$ is given in lines 258–259, where it is explicitly constrained by the condition $\mu_k \geq 1/(N\gamma)$. This threshold depends on the sample size $N$ and the step size $\gamma$. In the asymptotic regime where $N$ is very large, $k^*$ could indeed approach the full matrix dimension, and Theorem 4.1 then recovers the asymptotic behavior of learning algorithms. However, in the finite-sample regime—which is our primary focus- we have $k^∗<d$.
> - As mentioned in lines 834–835, we omitted the lower bound proof because it is structurally symmetric to the upper bound, differing only in the use of Theorem 5.2 from [2]. We apologize for missing the reference for Theorem 5.2 and will correct this in the revision. If helpful, we are happy to include the full proof of the lower bound in the revision.
>
> We appreciate the feedback and will revise the paper to clarify these points.
>
> Again, thank you so much for the detailed comments, questions and suggestions. They are really helpful for improving our paper, and we hope our response has addressed the concerns and questions you had.
>
> [1] "A convergence analysis of gradient descent on graph neural networks." Advances in Neural Information Processing Systems
>
> [2] “Benign overfitting of constant-stepsize sgd for linear regression”. In Conference on Learning Theory

---

### Official Review · Reviewer_mnhf · 2025-03-18

**Overall Recommendation:** 3

**Summary:**

This paper explores the interplay between graph structure and learning algorithms in Graph Neural Networks (GNNs), focusing on generalization performance (excessive risk) in the presence of noise. Extending learning theory to GNNs, it derives excessive risk profiles for Stochastic Gradient Descent (SGD) and Ridge regression, and linking these to graph structure via spectral graph theory. Key findings include: (1) SGD outperforms Ridge on power-law graphs, while Ridge excels on regular graphs; (2) stack GNNs introduce non-isotropic effects on excess risk, offering new insights into over-smoothing. Empirical results on synthetic graphs support the theory.

**Claims And Evidence:**

The claims are generally supported by clear and convincing evidence. The paper derives excess risk bounds (Theorems 4.1 and 4.2) for SGD and Ridge, connecting these to graph structure through spectral properties (Theorem 4.3), and extends the analysis to stacked GNNs (Proposition 4.4). These are backed by mathematical proofs and empirical validation using synthetic graphs.

**Essential References Not Discussed:**

I think the author conducted a relatively comprehensive literature review in both the paper and the appendix. However, some relevant papers on the over-smoothing problem in multi-layer GNNs could further enrich the discussion:

[1] Oono, K., & Suzuki, T. (2020). Optimization and generalization analysis of transduction through gradient boosting and application to multi-scale graph neural networks. Advances in Neural Information Processing Systems, 33, 18917-18930.

[2] Li, Qimai, Zhichao Han, and Xiao-Ming Wu. "Deeper insights into graph convolutional networks for semi-supervised learning." In Proceedings of the AAAI conference on artificial intelligence, vol. 32, no. 1. 2018.

**Experimental Designs Or Analyses:**

Overall experiments on synthetic datasets can support the theoretical findings to some degree.

However, as mentioned earlier, the paper lacks validation on real-world datasets, which could help identify practical limitations.

The author should also conduct a sensitivity analysis on graph structure hyperparameters, such as performance variations under different β for graph generation and different graph sizes.

**Methods And Evaluation Criteria:**

Overall, the methods are well-founded. Using synthetic graph models (e.g., Barabási-Albert) as benchmarks is reasonable for controlled theoretical validation, though incorporating real-world datasets could enhance practical relevance.

**Other Comments Or Suggestions:**

no

**Other Strengths And Weaknesses:**

Overall, this paper is well-written, with a clear organization and a logical proof structure that makes it easy to follow. The author effectively links graph structure and learning algorithms through spectral graph theory, deriving interesting insights that are validated by empirical studies.

As partially noted earlier and acknowledged by the authors—a commendable practice that should be encouraged—this paper also exhibits certain limitations:

(1) The idealized eigenvalue model (Eq. 4.1) may not adequately capture the complexity of real-world graph structures. Furthermore, the absence of evaluations on real-world datasets restricts the applicability of the theory to practical use cases.

(2) The theoretical analysis primarily focuses on the interplay between graph structure and learning algorithms, overlooking the role of node features. A future study jointly examining the interactions among all three—graph structure, learning algorithms, and node features—would be a valuable extension.

(3) The theory is limited to one-layer or linearized GNNs and does not extend to more complex models, which could broaden its scope and relevance.

**Questions For Authors:**

(1)	Equation 4.1 assumes a power-law decay (β). I am curious about how this parameter affects the results, as well as the impact of other hyperparameters, such as graph size.

(2)	How sensitive are your results to deviations from this model in real graphs? For example, some graphs sampled from social or internet data exhibit power-law characteristics, but they may also contain more noise. I would be interested in seeing an analysis of this.

**Relation To Broader Scientific Literature:**

Existing theoretical studies mainly focus on the convergence rates of learning algorithms in the interpolation regime (noise-free), providing only a coarse link between graph structure (e.g., maximum degree) and learning algorithms. In contrast, this paper offers a deeper exploration of how regular and power-law graphs influence SGD and Ridge, connecting their spectral properties to generalization bounds.

**Theoretical Claims:**

I reviewed the proofs for Theorems 4.1 (SGD excessive risk), 4.2 (Ridge excessive risk), 4.3 (graph structure effects), and Proposition 4.4 (multi-layer GNNs) in the main text and corresponding Appendix. To my knowledge, the proofs appear mathematically sound, though the reliance on idealized eigenvalue models (Eq. 4.1) could limit real-world applicability.

---

> ### Author Rebuttal · Authors · 2025-03-31
>
> Dear Reviewer, thank you for the positive comments and thoughtful questions! These greatly help us in making our paper better, and we appreciate the opportunity to address your concerns and questions here.
>
> ## Related Works in Over-smoothing
>
> Thank you for recommending relevant literature. We will incorporate the suggested references into our revision to enrich the discussion around the over-smoothing phenomenon in multi-layer GNNs.
>
> ## How the power-law decay $\beta$ and graph size affect the results
>
> - The parameter $\beta$ controls the rate of eigenvalue decay in the graph matrix spectrum. A larger $\beta $ results in a faster decay of the eigenspectrum, similar to that seen in power-law graphs, while a smaller $\beta$ leads to a more uniform eigen-spectrum, akin to that of regular graphs. We utilize $\beta$ to interpolate between these two regimes. Our theoretical results (Theorem 4.3) show that when $\beta$ is large(the graph exhibits stronger power-law behavior), SGD outperforms Ridge regression. Conversely,  when $\beta$ is small (the graph is more regular), Ridge regression becomes preferable.
> - Regarding the graph size $N$, it corresponds directly to the number of learning samples in our setting. Our analysis focuses explicitly on the generalization regime, requiring sufficiently large sample sizes for meaningful theoretical comparisons. Once this condition is met, increasing $N$ further does not affect the relative performance of SGD and Ridge—their comparison is primarily governed by the graph spectrum, not the graph size.
>
> ## Idealize eigenvalue model and sensitivity of results
>
> Thank you for the insightful question. Our theoretical results are not tightly dependent on the exact form of the power-law decay model. We adopt this model primarily for analytical clarity, but the core insights extend to a broader class of spectral decay behaviors.
>
> The essence of our findings (e.g., Theorem 4.3) lies in the rate at which the eigenvalues of the graph matrix decay, regardless of whether this follows a strict power-law. A faster-decaying spectrum—as often observed in graphs with power-law-like structures—tends to favor SGD. In contrast, a slower-decaying spectrum—as seen in more regular graphs—makes Ridge more favorable.
>
> Therefore, even in real-world graphs (e.g., social or internet networks) that only approximately follow a power-law and may contain additional noise or structural irregularities, our main theoretical tools (Theorems 4.1-4.2) remain applicable, and our theoretical conclusions regarding the relative performance of SGD and Ridge remain qualitatively valid. Extending our analysis to incorporate more flexible spectral models is an interesting direction for future work (please also see our response to the potential extension to the random graph model for Reviewer 8EPb).
>
> Thank you again for your diligent review, and we will incorporate the discussion above into the revision. We hope our response has satisfactorily addressed your questions and concerns.

---

### Decision · Program_Chairs · 2025-05-01

**Decision:**

Accept (poster)

**Comment:**

This paper investigates how graph structure influences the generalization of learning algorithms in GNN, with a focus on excess risk. The authors connects pectral properties of graphs to the learning behavior of stochastic gradient descent and Ridge regression. They analyze how graph topologyy impacts algorithm performance. Empirical results on synthetic and small real-world graphs support their findings.

Some reviewers noted that the scope of GNN models considered is limited. There are concerns about the realism of certain assumptions and the reliance on spectral properties derived from latent representations rather than directly from the graph topology.
Despite these limitations, the paper offers a solid theoretical contribution that links learning algorithms and graph structure. Its insights are valuable for understanding algorithm performance under different graph regimes and provide a foundation for future extensions. I recommend (weakly) acceptance.